# Bidirectional regulation of postmitotic H3K27me3 distributions underlie cerebellar granule neuron maturation dynamics

**Vijyendra Ramesh[1†], Fang Liu[2†], Melyssa S Minto[2], Urann Chan[2], Anne E West[1,2]***

[1]Molecular Cancer Biology Program, Duke University, Durham, United States; [2]Department of Neurobiology, Duke University, Durham, United States

**Abstract** The functional maturation of neurons is a prolonged process that extends past the mitotic exit and is mediated by the chromatin-dependent orchestration of gene transcription programs. We find that expression of this maturation gene program in mouse cerebellar granule neurons (CGNs) requires dynamic changes in the genomic distribution of histone H3 lysine 27 trimethylation (H3K27me3), demonstrating a function for this chromatin modification beyond its role in cell fate specification. The developmental loss of H3K27me3 at promoters of genes activated as CGNs mature is facilitated by the lysine demethylase and ASD-risk gene, Kdm6b. Interestingly, inhibition of the H3K27 methyltransferase EZH2 in newborn CGNs not only blocks the repression of progenitor genes but also impairs the induction of mature CGN genes, showing the importance of bidirectional H3K27me3 regulation across the genome. These data demonstrate that H3K27me3 turnover in developing postmitotic neurons regulates the temporal coordination of gene expression programs that underlie functional neuronal maturation.

**\*For correspondence:**
west@neuro.duke.edu

[†]These authors contributed equally to this work

## Editor's evaluation

The significance of the work is the choice of analyzing the coordination of chromatin modifications guiding cerebellar granule maturation, an important biological process that is uniquely suited for such an analysis. A strength of the work is their comprehensive findings revealing changes in specific chromatin modifications in promoters and additional regulatory elements that come into play during sequential stages of maturation. Authors also provide evidence for gene expression changes that correlate with the changes in chromatin status, using tools available in vivo and in cell culture. The paper was accepted because the authors satisfactorily addressed the concerns of both reviewers.

## Introduction

Cell identity is determined during differentiation by the coordinated actions of chromatin regulatory factors, which serve to establish the accessibility of gene regulatory elements that control the transcription of cell-type specific genes. Histone proteins play a key role in determining the architecture of chromatin, and specific post-translational modifications of the histones are tightly associated with the transcriptional state of a given locus or gene (*Hyun et al., 2017*). Among these modifications, histone methylation occurs on lysine and arginine residues of specific histones, and has roles in both gene activation and repression (*Jambhekar et al., 2019*), and acts to alter gene transcription by recruiting the site-specific binding of effector proteins that carry out specialized functions (*Daniel et al., 2005*). H3K27me3 is a repressive histone modification that is best known for its role during cell-fate

determination (*van Mierlo et al., 2019*). H3K27 methylation is deposited by the Polycomb repressive complex 2 (PRC2) lysine methyltransferases EZH1 and EZH2 (*O'Carroll et al., 2001*; *Voncken et al., 2003*). H3K27me3 can be enzymatically removed by the H3K27-specific lysine demethylases KDM6A (UTX) and KDM6B (JMJD3) (*Swigut and Wysocka, 2007*). In mice, null mutations in *Ezh2* result in early embryonic lethality (*O'Carroll et al., 2001*), *Kdm6a* knockouts show severely impaired mid-gestational development (*Welstead et al., 2012*), and *Kdm6b* knockouts die at birth (*Burgold et al., 2012*) suggesting the physiological importance of H3K27me3 regulation for mammalian embryonic development.

Neurons are born very early in their overall lifespan, and they undergo significant transcriptional and functional changes during the postmitotic stages of their developmental maturation. The enzymes that write and erase H3K27me3 remain expressed in neurons both during postnatal stages of development and in the adult brain (*Wijayatunge et al., 2014*; *Wijayatunge et al., 2018*), suggesting that they have functions beyond fate determination that remain to be understood. Conditional deletion of both *Ezh1* and *Ezh2* in neurons of the adult mouse causes a slow global loss of H3K27me3 as well as progressive neurodegeneration and death showing the importance of these enzymes in adult neurons (*von Schimmelmann et al., 2016*). One possibility is that H3K27me3 is required for fate maintenance. For example, deleting EED in developing postmitotic dopamine or serotonin neurons in vivo leads not only to the loss of H3K27me3 but also to reduced expression of the enzymes that define the trans-mitter phenotype of these neuromodulatory neurons (*Toskas et al., 2022*). Other studies suggest that H3K27me3 might be required in terminally differentiated neurons to persistently repress genes that have the potential to drive trans-differentiation to alternate cell fates (*Ferrai et al., 2017*; *von Schimmelmann et al., 2016*).

The global depletion of H3K27me3 in postmitotic neurons takes months to appear after the deletion of EZH1/2 (*von Schimmelmann et al., 2016*) revealing that this histone modification is remarkably stable across the bulk of the genome. Although histone replacement is a major mechanism of H3K27me3 removal in dividing cells (*Shpargel et al., 2014*), histone turnover is very slow in post-mitotic neurons (*Maze et al., 2015*). Interestingly, H3K27-selective lysine demethylases are widely expressed in the adult brain suggesting that these enzymes may have a particularly important role in the local regulation of histone methylation patterns in postmitotic cells (*Swahari and West, 2019*). Both *Kdm6a* and *Kdm6b* are expressed in neurons over the course of brain development, though only *Kdm6b* shows enhanced expression following exposure of cells to neuronal differentiation cues (*Burgold et al., 2008*; *De Santa et al., 2009*; *Jepsen et al., 2007*; *Wijayatunge et al., 2018*). Germ-line knockout of *Kdm6b* does not impair the gross architecture of the embryonic brain, but it does result in perinatal lethality caused by disruption of the functional maturation of a respiratory circuit that is required for proper breathing after birth (*Burgold et al., 2012*). This phenotype arises as a result of the lost demethylase function of Kdm6b, because survival cannot be rescued by transgenic expression of a demethylase-dead Kdm6b in the knockout mice (*Burgold et al., 2012*). These data are important because they suggest that demethylation mediated by Kdm6b is essential for the proper function of postmitotic neurons. Chromatin regulators are one of the major classes of genes associated with autism spectrum disorder (ASD) and notably *KDM6B* has been identified as a high-confidence ASD risk gene in humans (*Satterstrom et al., 2020*). *Kdm6b* haploinsufficiency in mice is associated with ASD-like impairments in sociability suggesting its importance for proper social network function in the adult brain (*Gao et al., 2022*), however, the underlying chromatin and transcriptional mechanisms of these Kdm6b mutation phenotypes remain to be elucidated.

We use CGNs to model neurodevelopment because they comprise more than 99% of all neurons (*Consalez et al., 2021*) and 85% of cells (*Altman and Bayer, 1997*) in the cerebellum, enabling the in vivo comparison of chromatin across CGN developmental states with limited contamination from other cell types. When purified and placed in ex vivo culture, CGN progenitors coordinately exit the cell cycle, minimizing heterogeneity of cell developmental stage across the population (*Fogarty et al., 2007*). In both preparations, gene transcription programs change over time coincident with changes in CGN biology (e.g. proliferation, migration, synapse formation, and synapse maturation *Frank et al., 2015*; *Sathyanesan et al., 2019*). We previously showed that loss of *Kdm6a/b* had no effect on postnatal morphogenesis of the cerebellum, demonstrating these enzymes are not required in vivo for the regulation of CGN progenitor proliferation (*Wijayatunge et al., 2018*). However, knockdown of *Kdm6b* in cultured CGNs impaired the developmental induction of genes

during the period of postmitotic neuronal maturation (*Wijayatunge et al., 2018*). Based on these data, we hypothesized that H3K27me3 may serve as a temporal gatekeeper of these maturation genes, delaying their activation beyond the immediate period of terminal differentiation to the neuronal fate.

Here to directly test the hypothesis that H3K27me3 regulation serves as a mechanism of neuronal maturation we determined the role of H3K27me3 dynamics in changing programs of gene transcription in postmitotic CGNs. We used ChIP-seq and a CUT&RUN time course study to identify the genomic location and timing of differential sites of H3K27me3 enrichment across the genome of fate-committed CGNs developing in vivo and in culture. We observed that H3K27me3 distributions continue to change even after fate-committed CGN progenitors exit the cell cycle, and that the loss of H3K27me3 at gene promoters is associated with the activation of genes induced as neurons mature. We also found a smaller subset of differential H3K27me3 peaks that occur at poised enhancers, which are dually marked by H3K27me3 and H3K4me1 and inversely correlated over time with H3K27ac enrichment. We used both genetic and pharmacological approaches to disrupt the action of H3K27me3 regulatory enzymes and show evidence of roles for both the lysine demethylase KDM6B and the lysine methyltransferase EZH2 in the induction of a postmitotic program of gene transcription that underlies functional maturation of CGNs. Taken together, our data advance our understanding of how programs of gene expression are temporally orchestrated by changing H3K27me3 distributions in the chromatin of maturing neurons.

## Results

### Cerebellar maturation involves dynamic, genome-wide changes in H3K27me3 distribution

To measure H3K27me3 dynamics during cerebellar maturation in vivo, we harvested cerebellar cortex from C57BL/6NCrl mice at postnatal days 7, 14, and 60 (*Figure 1A*). At P7, about half of all granule cells in the mouse cerebellum are proliferating granule neuron precursors (GNPs) whereas the rest are immature, postmitotic cerebellar granule neurons (CGNs). By P14 GNP cell division has ceased, and the cerebellum is composed of immature postmitotic CGNs that are migrating to the inner granule layer (IGL) where they receive synapses and fully mature by P60 (*Altman and Bayer, 1997*; *Frank et al., 2015*). Western blot of acid-extracted histones showed a significant global increase in total H3K27me3 levels from P7 to P14 that was maintained at P60 (*Figure 1B*), whereas total H3K27ac fell in a similar temporal pattern (*Figure 1—figure supplement 1A*). These data are consistent with the PRC2-dependent restriction of cell potential upon terminal differentiation (*van Mierlo et al., 2019*).

However, when we assessed the genomic location of H3K27me3 by ChIP-seq (*Figure 1C–F*; *Figure 1—figure supplement 1B*) we observed a more complex pattern. We found local losses as well as gains in H3K27me3 enrichment at different sites between developmental stages (*Figure 1C*). This was true even when comparing P14 with P60 (*Figure 1C*), demonstrating that bidirectional changes in H3K27me3 distributions occur during postmitotic stages of neuronal maturation, well after the period of terminal cell fate commitment has ended (*Figure 1—figure supplement 1C*). We stratified differential H3K27me3 ChIP-seq peaks by their genomic location with respect to proximal gene promoters, gene bodies, or intergenic regions (*Figure 1D*), and found a consistent overrepresentation of proximal gene promoters among developmentally demethylated peaks for all three comparisons. By contrast, H3K27me3 ChIP-seq peaks differentially gained between all three comparisons did not differ from the global distribution of H3K27me3, which was distributed across gene-body and distal intergenic regions as well as proximal promoters. The percentage of total H3K27me3-positive promoters gradually decreased by ~ twofold from P7 to P60, while that of H3K27me3-positive distal intergenic regions increased, and that of gene bodies stayed about the same over time (*Figure 1E*). Additionally, the total number of H3K27me3 peaks called centered within a 5 kb window around the TSS of genes annotated with H3K27me3 appeared to gradually decrease from P7 to P60 (*Figure 1E*). These data demonstrate that in CGNs H3K27me3 patterns are locally regulated in postmitotic neurons, suggesting a function for H3K27me3-dependent gene regulation in developing neurons that extends beyond the restriction of cell fate commitment.

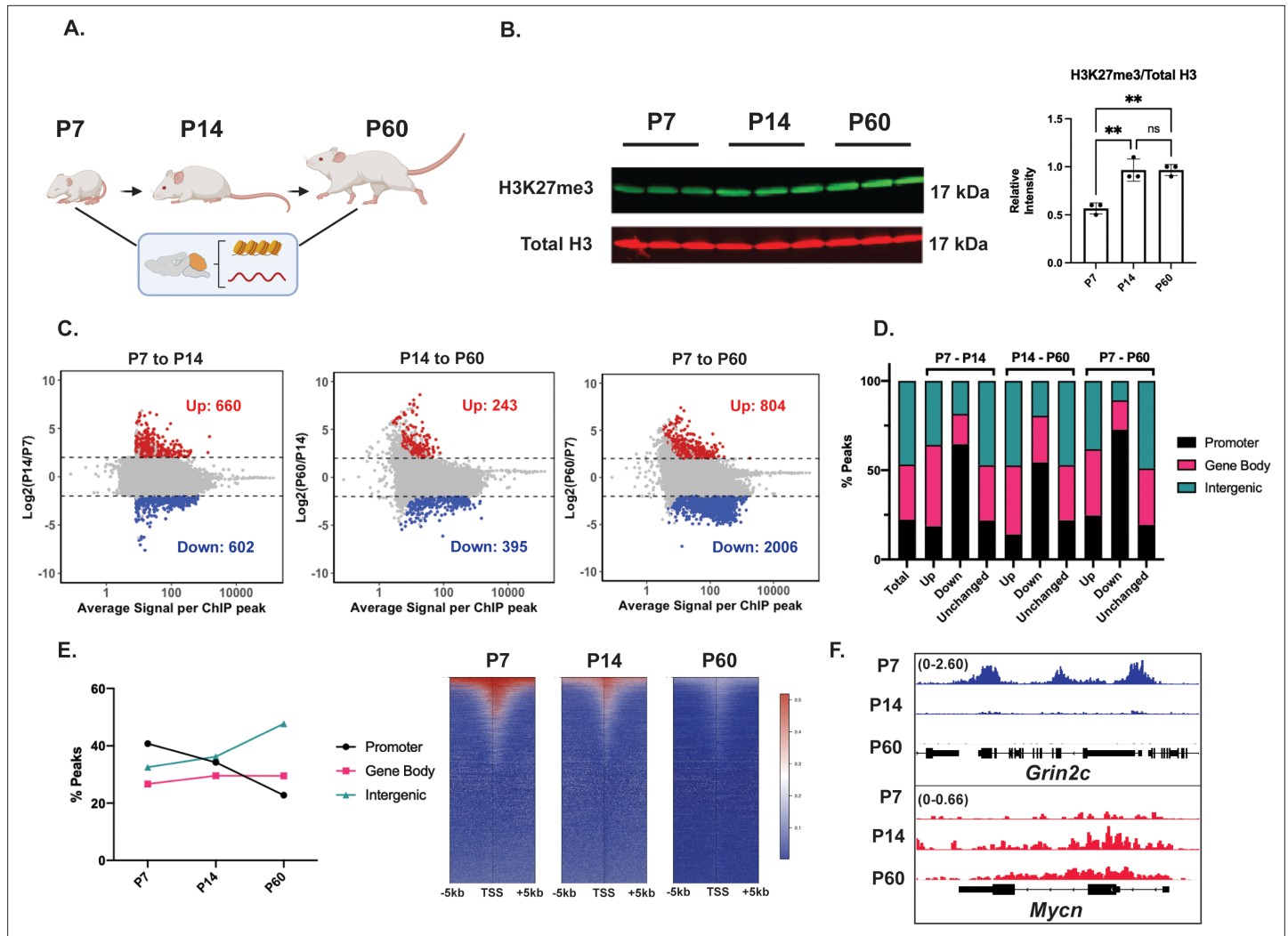

**Figure 1.** The maturing mouse cerebellum undergoes a genome-wide redistribution of histone H3 lysine 27 trimethylation (H3K27me3). (**A**) Mice were sacrificed at postnatal days 7, 14, and 60 after which cerebellar tissue was dissected out and processed to harvest histones, chromatin, and RNA. (**B**) (Left) Western blot of acid extracted histone from cerebellar tissue for H3K27me3 and total Histone H3 (n=3 biological replicates), (Right) Quantification of Western Blot, one-way ANOVA, ** indicates p<0.005. (**C**) MA plots showing H3K27me3 peaks gained (Up) and lost (Down) throughout cerebellar maturation in vivo (n=3 biological replicates, differential enrichment calculated using DESeq2 package. Up and Down are loci with FDR <0.05 and |L2FC|>2). (**D**) Percentage of differential H3K27me3 peaks between P7-P14, P14-P60, and P7-P60, annotated by genomic region. Annotation performed using ChIPseeker package, TSS +/− 3000 bp; 'Total' represents consensus H3K27me3 peaks across P7, P14, and P60. (**E**) (Left) Percentage of H3K27me3 peaks annotated by genomic region, (Right) Heatmap of H3K27me3 peaks centered around genes with H3K27me3 peaks TSS +/− 5000 *bp*. (**F**) ChIP-seq tracks for example 'Down' gene *Grin2c* (Upper) and 'Up' gene *Myc* (lower). The numbers at the top of each track indicate the y-axis scale, which is fixed for all tracks in the same set. Source Data.

The online version of this article includes the following source data and figure supplement(s) for figure 1:

**Source data 1.** Uncropped annotated blot of the image depicted in *Figure 1B* (short exposure).

**Source data 2.** Uncropped annotated blot of the image depicted in *Figure 1B* (long exposure).

**Figure supplement 1.** Histone H3 lysine 27 trimethylation (H3K27me3) ChIP-seq and RNA-seq profile of the developing cerebellum in vivo.

**Figure supplement 1—source data 1.** Uncropped annotated blots of the image depicted in *Figure 1—figure supplement 1A* (short exposure).

**Figure supplement 1—source data 2.** Uncropped annotated blots of the image depicted in *Figure 1—figure supplement 1A* (long exposure).

## H3K27me3 dynamics at promoters during cerebellar maturation inversely correlate with H3K4me3 enrichment and gene expression

We identified 3618 differential H3K27me3 peaks across all time points using a strict cut-off of |Log$_2$FC|>2, p-adj <0.05. We first filtered these for promoters because these regions can be directly

mapped to their target gene for transcriptional regulation and obtained 1805 peaks. We then asked whether peaks that were up- and downregulated over stages of CGN differentiation were found at genes that can be stratified according to cellular function. Examples of genes with promoter-associated peaks lost or gained from P7 to P60, respectively, were the genes encoding the mature NMDA-type glutamate receptor *Grin2c* which is strongly expressed in mature CGNs (*Wijayatunge et al., 2018*) and the cell-proliferation factor *Mycn,* which is expressed only in GNPs (*Ma et al., 2015*; *Figure 1F*). This is consistent with the hypothesis that changes in the levels of promoter H3K27me3 inversely correlate with changes in gene expression over development.

To determine if the correlation of H3K27me3 demethylation with transcriptional maturation holds broadly, we first performed hierarchical clustering on the 1805 differential H3K27me3 promoter peaks and identified three major clusters with differing direction and timing of change (*Figure 2A*). The first cluster, which we termed 'H3K27me3 Up,' contained peaks that gained methylation as early as P14 and maintained methylation during CGN maturation. The next two clusters included demethylated peaks, that we termed 'H3K27me3 Down (Fast)' and 'H3K27me3 Down (Slow)', respectively, depending on whether they changed between P7 and P14 or between P14 and P60. Because H3K27me3 at promoters has been associated with H3K4me3 bivalency during cell-fate specification (*Bernstein et al., 2006*), a state that is thought to poise genes for later activation, we next asked whether the presence of H3K27me3 at these gene promoters was associated with H3K4me3. We downloaded H3K4me3 ChIP-seq data from P6 and P22 mouse cerebellar tissue (*Yamada et al., 2014*), and compared both the presence and the levels of H3K4me3 across development at promoters of genes within the differential H3K27me3 clusters (*Figure 2A*). Regardless of their H3K27me3 dynamics, most of these promoters had peaks of H3K4me3 at both P6 and P22 (*Figure 2B*). However, these data also show that, compared with the set of promoters that have unchanged H3K27me3 over development, there was a significant decrease in H3K4me3 levels over time at promoters that gain H3K27me3, whereas there is a significant increase in H3K4me3 at promoters that lose H3K27me3 (*Figure 2B*). Further consistent with the hypothesis that turnover of promoter H3K27me3 permits promoter activation, we saw an inverse correlation between changes in H3K27me3 and enrichment of the active chromatin marker H3K27ac as well as chromatin accessibility (*Frank et al., 2015*) at the same promoters (*Figure 2—figure supplement 1*).

Gene Ontology (GO) analysis showed the H3K27me3 Up category to be enriched for genes associated with progenitor functional terms related to cell proliferation and development, whereas the H3K27me3 Down (Fast) and Down (Slow) clusters were both enriched for neuronal GO terms (*Figure 2C*). Because developmentally gained peaks showed a broader distribution than just promoters, we performed the same type of filtering and hierarchical clustering for differential H3K27me3 peaks overlying gene bodies (*Figure 2—figure supplement 2*). Again, these data show that gene bodies that lose H3K27me3 are associated with terms related to genes expressed in mature neurons, however, these are much fewer in number compared to promoters. To determine if H3K27me3 dynamics at promoters were associated with the expression of the corresponding RNA transcript across CGN maturation (*Figure 2D*), we compared our H3K27me3 ChIP-seq to a dataset of gene expression at the same time points from our previous study (*Frank et al., 2015*). We found that 'H3K27me3 Up' peaks at gene promoters were significantly associated with a decrease in expression of the associated gene over the same period of developmental time and 'H3K27me3 Down' peaks were associated with an increase in gene expression. Among the H3K27me3 Down peaks, 'Fast' peaks were associated with a significantly greater magnitude of gene induction than 'Slow' peaks between P7 and P14, but 'Slow' peaks were associated with a significantly greater magnitude of gene induction than 'Fast' peaks between P14 and P60. There was no significant difference between 'Fast' and 'Slow' peaks from P7 to P60 indicating that it is the timing not the magnitude of differential developmental gene expression that defines the two clusters. These data show that the H3K27me3 dynamics at gene promoters are associated with differential H3K4me3 enrichment, and the timing of gene expression changes in postmitotic neurons, which allows for multiple postmitotic waves of gene expression (*Figure 2E*). Taken together, these data provide evidence that the postmitotic removal of H3K27me3 from a subset of gene promoters is important for the coordinated induction of gene expression that underlies neuronal maturation.

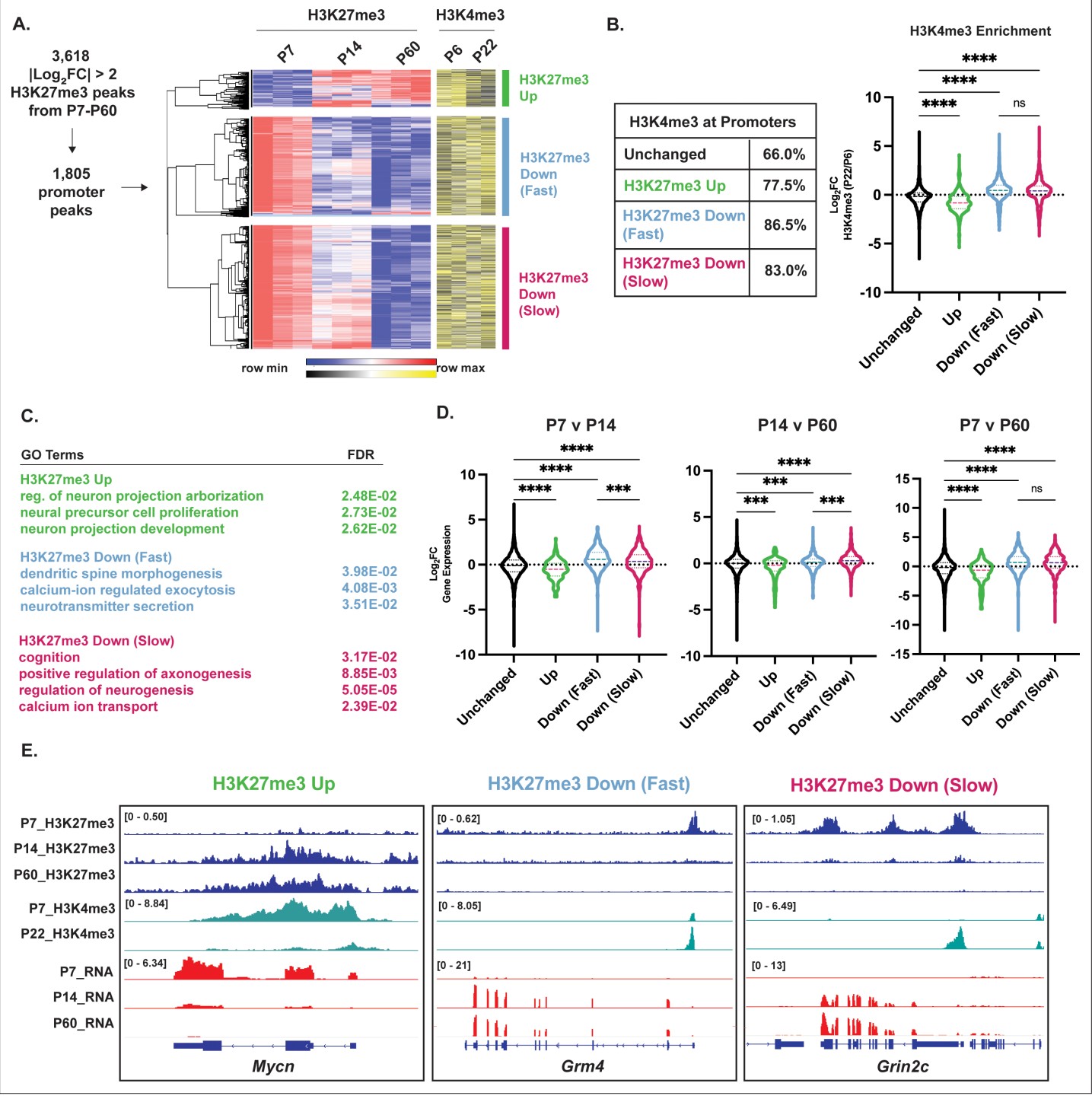

**Figure 2.** Differential histone H3 lysine 27 trimethylation (H3K27me3) promoters cluster by turnover kinetics and correlate with cerebellar granule neuron (CGN) maturation gene expression. (**A**) Differential H3K27me3 peaks across P7 to P60 were filtered for promoter-associated peaks, followed by the hierarchical clustering with Pearson correlation, of their VST-transformed DESeq2-normalized counts (left) from P7-P60, and corresponding VST-transformed DESeq2-normalized counts for H3K4me3 from P6-P22 (adapted from *Yamada et al., 2014*). (**B**) (left) Table showing percent overlap of promoters in clusters described in A with promoters with H3K4me3. (right) Violin plot showing the distribution of Log2FC of H3K4me3 (P22/P6) as a function of clustering performed in A, (one-way ANOVA, **** indicates p<0.0001). (**C**) Gene Ontology (GO) Terms associated with the nearest gene and FDR for clusters described in A. (**D**) Violin plot showing the distribution of Log2FC of gene expression, measured by RNA-seq as a function of clustering performed in A, and genes with unchanged H3K27me3 across developmental time (one-way ANOVA, **p<0.005, ***p<0.001, ****p<0.0001). (**E**) Example H3K27me3 (blue) and H3K4me3 (green) ChIP-seq and RNA-seq (red) tracks belonging to CGN maturation genes within clusters described in A, at *Mycn*, *Grm4,* and *Grin2c* loci. The numbers at the top of each track indicate the y-axis scale, which is fixed for all tracks in the same set.

*Figure 2 continued on next page*

*Figure 2 continued*

The online version of this article includes the following figure supplement(s) for figure 2:

**Figure supplement 1.** Histone H3 lysine 27 trimethylation (H3K27me3) removal is followed by a gain of H3K27ac and chromatin accessibility at gene promoters.

**Figure supplement 2.** Differential histone H3 lysine 27 trimethylation (H3K27me3) enrichment at gene body regions and corresponding GO terms.

## H3K27me3 is bidirectionally regulated at a sparse set of cerebellar enhancers

Widespread changes in the activation and function of key developmentally regulated enhancers have been identified during cerebellar maturation in vivo over embryonic (*Ramirez et al., 2022*) and post-natal (*Frank et al., 2015*) timepoints. To determine whether the non-promoter H3K27me3 peaks we identified in developing cerebellum might function to regulate enhancers, we assessed our differential H3K27me3 sites for histone markers that are characteristic of enhancers. Like the concept of bivalent promoters, the dual presence of H3K4me1 and H3K27me3 at enhancers is proposed to poise them for future activation (*Rada-Iglesias et al., 2011*). Thus, we first computed the overlap of the

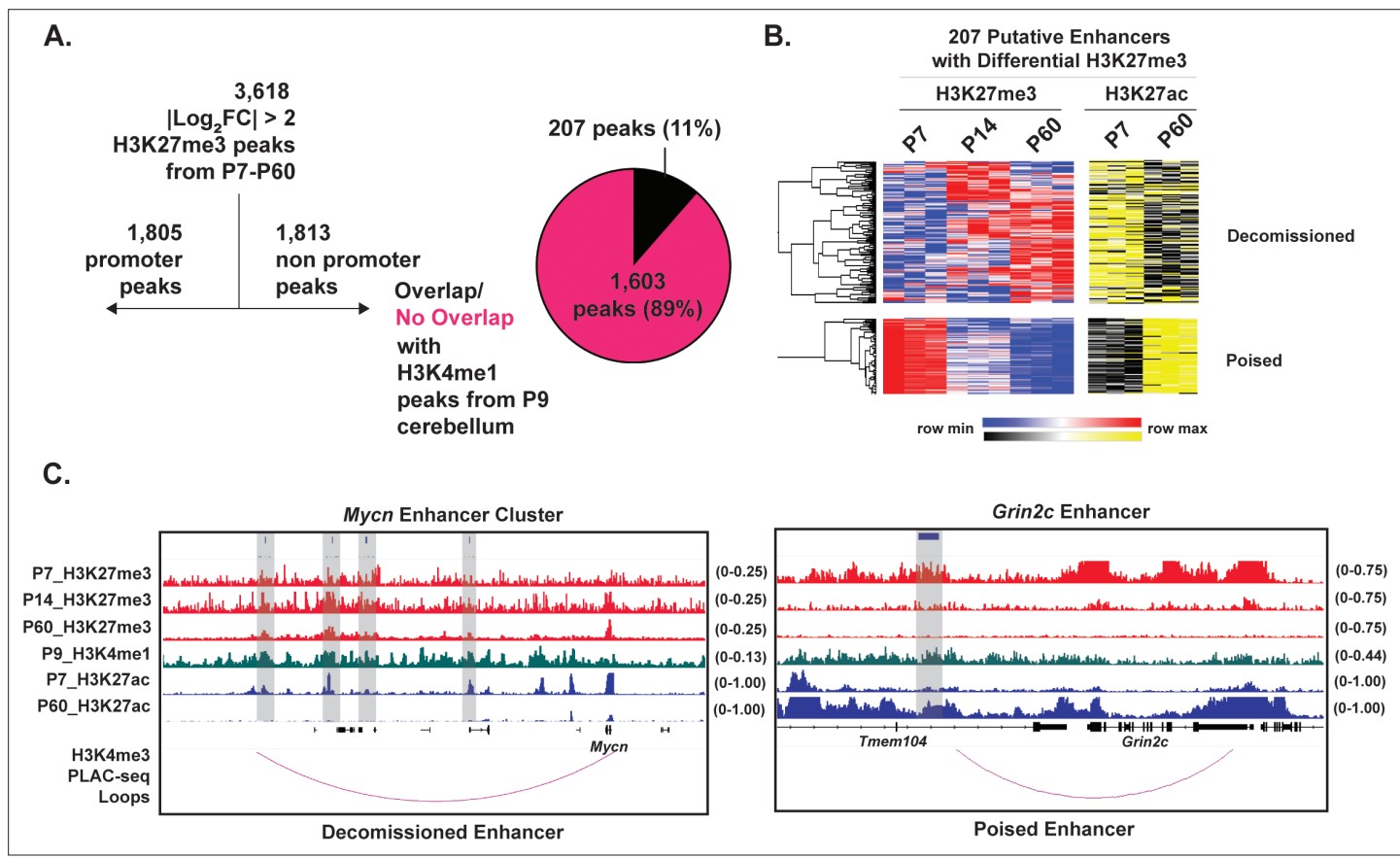

**Figure 3.** Histone H3 lysine 27 trimethylation (H3K27me3) is bidirectionally regulated at a sparse set of developmentally regulated enhancers. (**A**) Filtering for differential H3K27me3 peaks from P7-14-60 with |Log2FC|>2, for non-promoters (TSS +/− 3 kb), followed by overlap with H3K4me1 peaks obtained from P9 cerebellum from *Ramirez et al., 2022*, (n=2) biological replicates. (**B**) Heatmap of spearman rank correlated, hierarchically clustered, VST-transformed DESeq2-normalized counts of differential H3K27me3 peaks at putative enhancers (left) from P7-P60, and corresponding VST-transformed DESeq2-normalized counts for H3K27ac from P7-P60 adapted from *Frank et al., 2015*. (**C**) (Left) Tracks for H3K27me3 (red), H3K4me1 (green) and H3K27ac (blue) and H3K4me3 (purple) PLAC-seq loops from *Yamada et al., 2019* at putative enhancer for *Mycn*. (**D**) Same as 'C' but for putative *Grin2c* enhancer. Gray bars indicate putative enhancers. The numbers at the top of each track indicate the y-axis scale, which is fixed for all tracks in the same set.

The online version of this article includes the following figure supplement(s) for figure 3:

**Figure supplement 1.** Only a subset of differential H3K27ac peaks at non-promoters is also differentially H3K27-trimethylated.

1813 differential non-promoter H3K27me3 peaks with H3K4me1 peaks derived from P9 cerebellum (*Ramirez et al., 2022*) and found 207 putative co-labeled enhancers (*Figure 3A and B*). To determine if the developmental change in H3K27me3 at these regulatory elements was correlated with a change in their function, we determined the relationship between developmental changes in H3K27me3 and changes in the levels of H3K27ac, which is widely used as a marker of enhancer activity (*Ernst and Kellis, 2013*). As we did in our analysis of H3K27me3 at promoters, we first clustered H3K27me3 peaks at putative enhancers by their H3K27me3 levels over time and identified two clusters that increase or decrease H3K27me3 as CGNs mature (*Figure 3B*). These peak clusters were inversely associated with changes in H3K27ac levels over developmental time, such that putative enhancers gaining H3K27me3 by P60 lost activity (we term these 'Decommissioned') and putative enhancers losing H3K27me3 by P60 gained activity (we term these 'Poised'). Within the 'Decommissioned' cluster, we identified 4 putative enhancers near *Mycn* that were also identified in the 'Developing Mouse Cerebellum Enhancer Atlas' (*Ramirez et al., 2022*). Within the 'Poised' cluster, we identified an enhancer near *Grin2c* in a region we previously showed regulates *Grin2c* expression (*Frank et al., 2015*). Both of these putative enhancers physically interact with the promoters of their putative target genes as measured by H3K4me3 PLAC-seq (*Goodman et al., 2020*) consistent with them playing a functional role in transcription.

These data show that although the vast majority (3411/3618; ~95%) of differential H3K27me3 peaks in developing mouse cerebellum do not overlap enhancers with H3K4me1, regulated changes in H3K27me3 are likely to be functionally important at a subset of these enhancers for controlling the expression of developmentally regulated genes. As an alternative way to determine the extent of H3K27me3 regulation at cerebellar enhancers, we started with the full set of non-promoter sites that showed differential regulation of H3K27ac over cerebellar development as our catalog of developmentally regulated enhancers (*Frank et al., 2015*) and then asked how many of these sites overlapped H3K27me3 (*Figure 3—figure supplement 1A*). We identified 13,409 differential H3K27ac peaks that were not at promoters (*Figure 3—figure supplement 1B*). However, only 214 of these, about 1.6%, also showed developmentally regulated changes in H3K27me3. Thus although enhancers are crucially important regulatory elements for controlling cell type- and developmental stage-specific regulation of gene transcription (*Nord and West, 2020*), the developmental changes in H3K27me3 at CGN regulatory elements appears to be biased toward gene promoters.

## *Kdm6b*-cKO impairs CGN maturation via promoter hypermethylation

The developmental loss of H3K27me3 in CGNs could occur through changes in the expression of the enzymes that maintain or remove these histone modifications. At the level of gene expression, RNA-seq counts of the H3K27 methyltransferases *Ezh1/2* and the H3K27 demethylases *Kdm6a/b* showed that although there was a developmental switch between the dominant reader and writer enzymes, the overall expression of each class of enzyme did not substantially change over time (*Figure 4—figure supplement 1*). However, previous in situs from our lab demonstrated that *Kdm6b* expression is transiently induced in newborn CGNs in the inner EGL, and we further showed that shRNA-mediated knockdown of *Kdm6b* in cultured CGNs impairs the expression of a subset of developmentally induced genes (*Wijayatunge et al., 2018*). Thus, we considered the possibility that KDM6B might mediate the postmitotic loss of H3K27me3 at a subset of developmentally induced genes in the developing cerebellum in vivo to promote their transcriptional induction.

To test this hypothesis, we conditionally knocked out *Kdm6b* in cerebellar GNPs in vivo by crossing heterozygous transgenic *Atoh1-Cre* mice with *Kdm6b^fl/fl^* mice. We then harvested cerebellar tissue from littermates that were either *Kdm6b^fl/fl^* (WT) or *Atoh1-Cre; Kdm6b^fl/fl^* (*Kdm6b*-cKO) at P14 as an intermediate developmental time point for CGN maturation (*Frank et al., 2015*; *Figure 4A*). We processed this tissue to perform either RNA-seq (*Figure 4—figure supplement 2A*) or ChIP-seq for H3K27me3 (*Figure 4B*; *Figure 4—figure supplement 2B*). We found 111 genes that had significantly elevated expression in the *Kdm6b*-cKO and 150 that showed significantly reduced expression (*Figure 4—figure supplement 2A*). Genes that were elevated at P14 in the *Kdm6b*-cKO mice all showed developmentally regulated expression in control mice that were highest at P7 and fell by P60. Genes whose expression was reduced in the *Kdm6b*-cKO mice also showed developmentally regulated expression, but they increased in expression over developmental time. Thus, the deletion of Kdm6b in GNPs appears to slow the transcriptional maturation of CGNs in postnatal life.

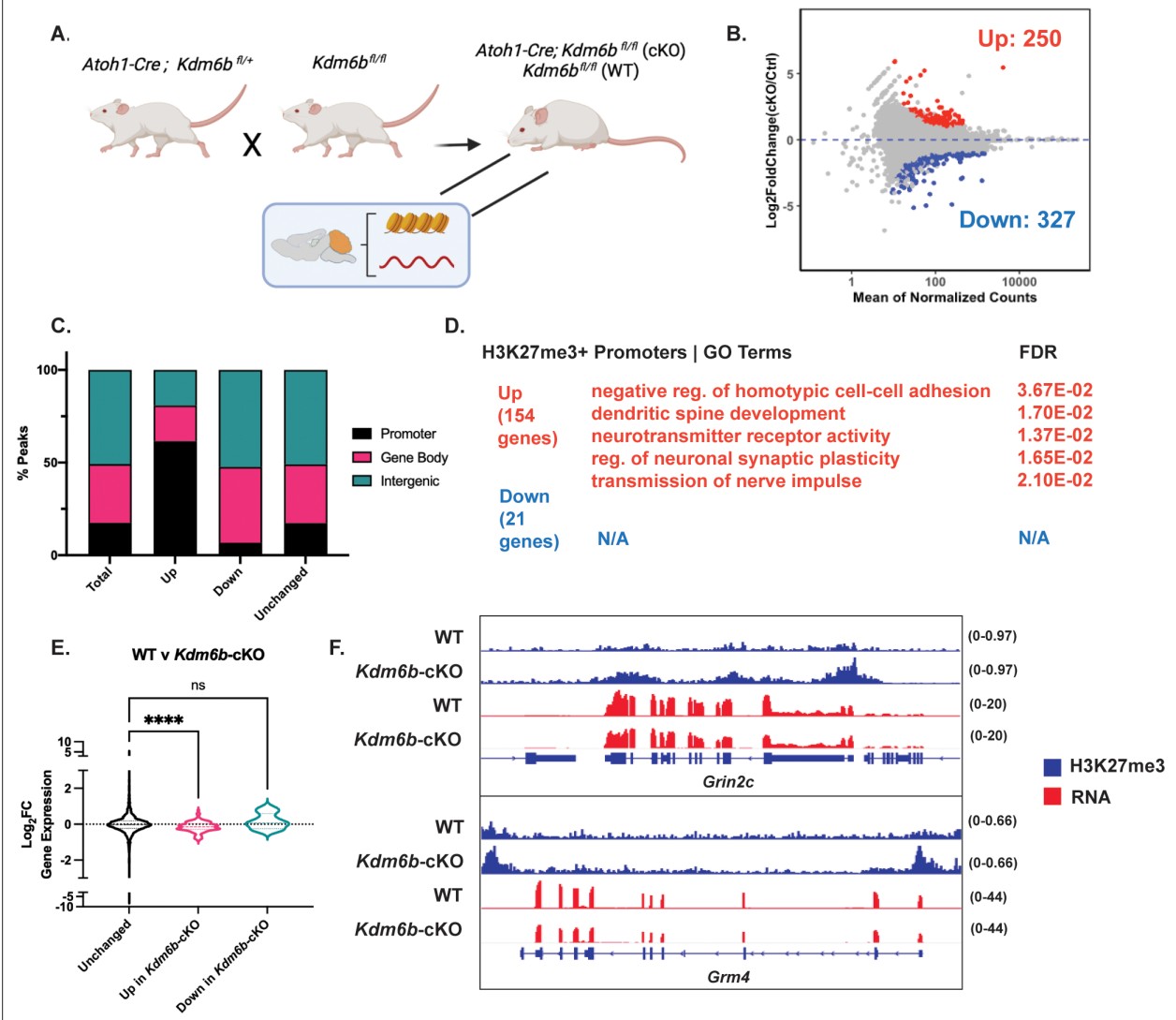

**Figure 4.** *Kdm6b* knockout in cerebellar granule neuron (CGN) precursors impairs CGN maturation via histone H3 lysine 27 trimethylation (H3K27me3) hypermethylation of CGN maturation gene promoters. (**A**) *Atoh1-Cre; Kdm6b*$^{fl/fl}$ (cKO) and *Kdm6b*$^{fl/fl}$ (WT) mice were generated by crossing *Atoh1-Cre; Kdm6b*$^{fl/+}$ mice to *Kdm6b*$^{fl/fl}$ mice and were sacrificed at P14 for cerebellar tissue harvest and subsequent performing of H3K27me3 ChIP-seq (WT n=2, cKO n=3 biological replicates) and RNA-seq (n=2 biological replicates). (**B**) MA plot showing differential enrichment of genome-wide H3K27me3 peaks between WT and cKO mice (|Log2FC|>1, FDR <0.05). (**C**) Differentially methylated H3K27me3 between WT and cKO annotated by genomic location using ChIPseeker package, TSS +/− 3000 bp. (**D**) Gene Ontology (GO) Terms and FDR for genes nearest gene promoters within H3K27me3 Up and Down (due to *Kdm6b*-cKO) peaks. (**E**) Distribution of Log$_2$Fold Change Expression (WT v cKO, computed using DESeq2) as a function of unchanged, H3K27me3 Up and H3K27me3 Down genomic promoters between WT and cKO (one-way ANOVA, **** indicates p<0.0001). (**F**) Representative genes containing H3K27me3 (blue) Up genomic promoter peaks *Grin2c* and *Grm4* and corresponding RNA-seq (red) tracks between WT and cKO mice. The numbers to the right of each track indicate the y-axis scale, which is fixed for all tracks in the same set.

The online version of this article includes the following figure supplement(s) for figure 4:

**Figure supplement 1.** Expression of histone H3 lysine 27 trimethylation (H3K27me3) Writers and Erasers in vivo.

**Figure supplement 2.** *Kdm6b*-cKO in *Atoh1-Cre*+ granule neuron precursors (GNPs) Impairs cerebellar granule neuron (CGN) Maturation in vivo.

We used DESeq2 to compute differentially enriched H3K27me3 ChIP-seq peaks between WT and *Kdm6b*-cKO tissue at P14, and obtained 250 upregulated peaks, and 327 downregulated peaks (FDR <0.05, |Log2FC|>1) (*Figure 4B*). We then mapped these differential peaks to their genomic location (*Figure 4C*). We found peaks with elevated levels of H3K27me3 in *Kdm6b*-cKO mice to be significantly more likely to be within gene promoters compared with the overall distribution of peaks or the distribution of peaks that had reduced H3K27me3 in *Kdm6b*-cKO mice. We observed that the

percentage of promoter-associated H3K27me3 peaks decreased between WT and *Kdm6b*-cKO mice, while peaks in distal intergenic regions and gene-bodies increased (***Figure 4—figure supplement 2C***). The genes containing promoters with increased H3K27me3 in the *Kdm6b*-cKO were enriched for neuronal GO terms (***Figure 4D***), whereas those containing promoters with decreased H3K27me3 in the *Kdm6b*-cKO mice failed to pull any significant GO terms. Overall, these observations mirror our findings of H3K27 demethylation at the promoters of neuronal maturation genes and suggest that KDM6B mediates that process.

To determine the relationship between H3K27me3 and gene expression in the *Kdm6b*-cKO mice, we computed the correlation between *Kdm6b*-cKO associated differential H3K27me3 enrichment at promoter regions, with the expression of the associated gene in WT and *Kdm6b*-cKO mice. These data showed that the gain of H3K27me3 is associated with a significant loss of gene expression (***Figure 4E***) as exemplified by the synaptic glutamate receptor subunit genes *Grin2c* and *Grm4* (***Figure 4F***). By contrast, genes that lost H3K27me3 at their promoters in *Kdm6b*-cKO mice showed no significant difference in gene expression (***Figure 4E***). These data suggest that developmental loss of H3K27me3 at the promoters of genes induced in CGNs during postmitotic maturation is facilitated by KDM6B, and that this H3K27me3 removal promotes the developmental induction of gene expression.

## The ZIC transcription factors are enriched at promoters that show developmental loss of H3K27me3

To determine what characterizes the subset of promoters that become demethylated we sought to ask what transcription factors might be found in these gene promoters compared to all H3K27me3-positive

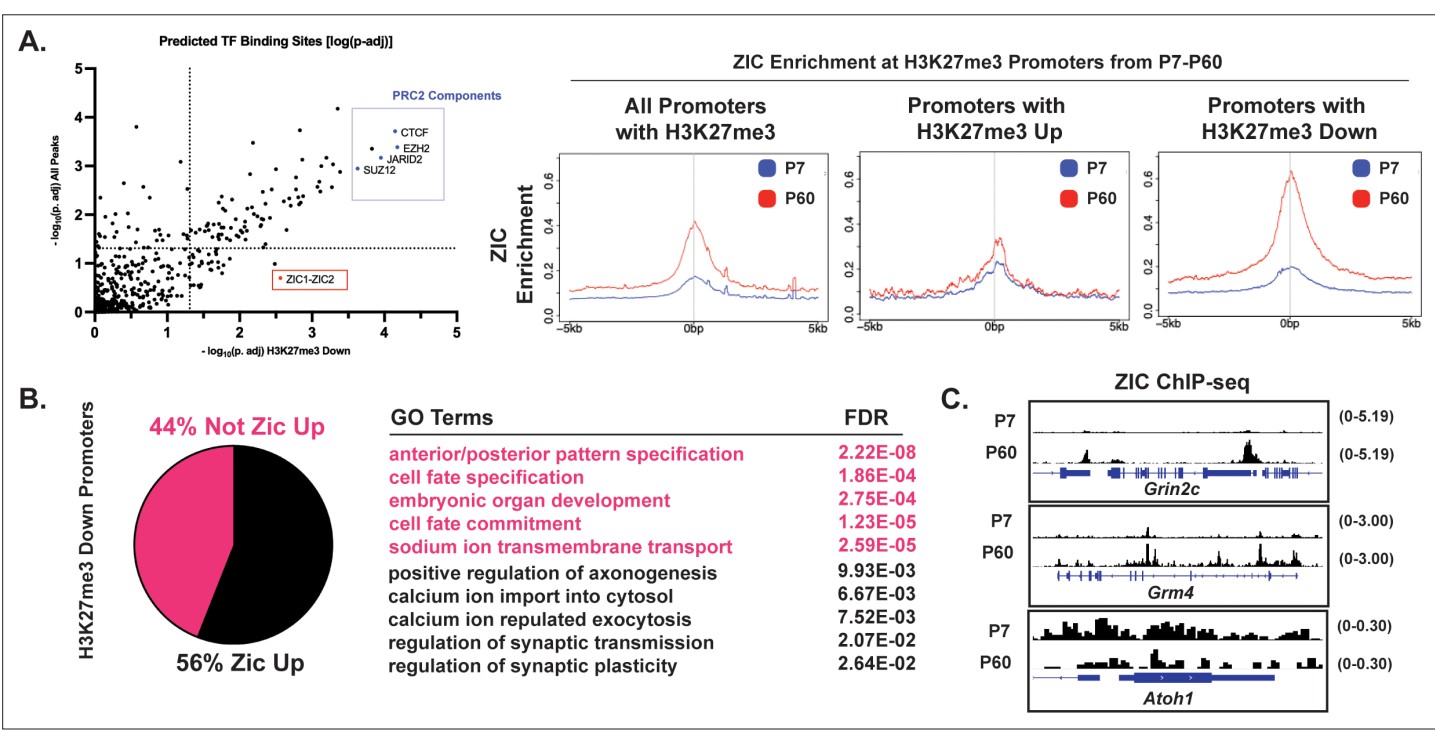

**Figure 5.** Histone H3 lysine 27 trimethylation (H3K27me3) removal is accompanied by the gain of ZIC at gene promoters. (**A**) (Left) Binding Analysis for Regulation of Transcription (BART) for genes near promoters within H3K27me3 Down peaks (x-axis) and near all H3K27me3 peaks (y-axis) plotted by -log$_{10}$p-adj. (Right) Metagene plots for ZIC ChIP-seq signal at genes near all promoters with H3K27me3, promoters with H3K27me3 Up, and promoters with H3K27me3 down. (**B**) (Left) Percentage of genes near H3K27me3 Down promoters that are also ZIC1/2 Up (Log$_2$Fold Change >0, P60/P7, ZIC1/2 ChIP-seq). (Right) Gene Ontology (GO) Terms and FDR for genes nearest to H3K27me3 Down promoter peaks that are also ZIC1/2 Up, and not ZIC1/2 Up. (**C**) ZIC ChIP-seq tracks at P7 and P60 at H3K27me3 Down promoters of Grin2c and Grm4, compared to the early gene Atoh1. The numbers to the right of each track indicate the y-axis scale, which is fixed for all tracks in the same set.

The online version of this article includes the following figure supplement(s) for figure 5:

**Figure supplement 1.** Histone H3 lysine 27 trimethylation (H3K27me3) turnover at CGN-maturation gene promoters is associated with gain of ZIC binding.

promoters. To conduct an unbiased screen, we first utilized the tool called Binding Analysis for Regulation of Transcription (BART) (*Wang et al., 2018*), which calculates enrichments of previously published ChIP-seq TF binding profiles within a defined set of genomic regions. We first filtered our ChIP-seq data for genes associated with the 'H3K27me3-Down' peak cluster described in *Figure 2A*, combining 'Down (Fast),' and 'Down (Slow)' clusters, and obtained predicted transcription factors (TF) whose binding significantly overlaps these gene promoters. We compared the enrichment statistics of TFs enriched in these clusters to those derived from all H3K27me3– promoter-associated genes (*Figure 5A*). As expected, we found PRC2 components EZH2, JARID2, and SUZ12 to be significantly enriched across both the H3K27me3-Down cluster and all H3K27me3-promoter-associated genes. However, we found only the H3K27me3-Down cluster to be significantly enriched for the putative binding of transcription factors ZIC1 and ZIC2, which previous work from our group has shown to be a driver of CGN maturation (*Frank et al., 2015*). Importantly, when we assessed the 'H3K27me3-Down (Fast),' and 'H3K27me3-Down (Slow)' clusters separately (*Figure 5—figure supplement 1A*), they were also enriched for the binding of ZIC1 and ZIC2, which is consistent with evidence (*Frank et al., 2015*) that these TFs function at multiple stages of CGN differentiation. By contrast, when we performed the same analysis for the 'H3K27me3-Up' cluster (*Figure 5—figure supplement 1A*), we found this set of promoters was significantly enriched for the binding of a distinct set of TFs involved in cell differentiation, namely SOX3, FOXF1, SHOX2, POU3F2 as well as histone deacetylation (HDAC1) (*Wang et al., 2006*).

To directly compare changes in ZIC1/2 TF binding to changes in H3K27me3 across CGN development, we re-analyzed our previously published ZIC1/2 ChIP-seq data (*Frank et al., 2015*) and measured ZIC1/2 binding at promoters of All Genes, Genes with H3K27me3 Up, and Genes with H3K27me3 Down (*Figure 5A*). We found ZIC1/2 enrichment to increase most notably at Genes with H3K27me3 Down, compared to All Genes and Genes with H3K27me3 Up. We then identified gene promoters with increased ZIC binding at P60 compared with P7, $|Log_2FC|>0$, p-adj <0.05. We termed these genes 'ZIC Up' and computed their overlap with genes within the H3K27me3 Down cluster. We found that 56% of H3K27me3-Down genes were also ZIC-Up, and that these genes were strongly enriched for neuronal GO terms (*Figure 5B*), while the 44% of genes that were not ZIC-Up were enriched for generic cell-fate specification GO terms. As a control, we compared 'ZIC Up' genes to H3K27me3-Up genes (*Figure 5—figure supplement 1A*) and only found an overlap of only about 11%. We repeated this analysis with the H3K27me3 ChIP-seq peaks that were differential at P14 between WT and *Kdm6b*-cKO described in *Figure 4* and found a similar enrichment of putative ZIC1 and ZIC2 binding within gene promoters associated with H3K27me3 peaks 'Up in *Kdm6b*-cKO' (*Figure 5—figure supplement 1B*), but not 'Down in *Kdm6b*-cKO' (*Figure 5—figure supplement 1C*). 72% of the 'H3K27me3 Up in *Kdm6b*-cKO' genes were also 'ZIC Up' (*Figure 5—figure supplement 1B*) and were strongly enriched for neuronal GO terms. Only 9% of 'H3K27me3 Down in *Kdm6b*-cKO' were also 'ZIC Up' (*Figure 5—figure supplement 1C*). As examples of gene promoters that belong to both the H3K27me3-Down cluster, and H3K27me3 Up peaks due to *Kdm6b*-cKO, we show tracks for *Grin2c* and *Grm4,* which are strongly enriched for ZIC1/2 at their promoters at P60 (*Figure 5C*).

## H3K27me3 turnover dynamics are associated with CGN gene maturation in culture

To determine the functional consequences of modulating H3K27me3 we characterized cultures of primary mouse CGNs as a context for pharmacological manipulation of histone regulatory enzymes. These CGNs were derived from GNPs isolated from the P6-P8 mouse cerebellum and were grown in vitro for up to 7 days (7DIV). Prior studies have shown that GNPs exit the cell cycle by 1DIV (*Fogarty et al., 2007*). To measure the concordance of gene expression changes between cerebellar maturation in vivo and in culture, we performed a rank-rank hypergeometric overlap (RRHO) analysis of gene expression over time in the two contexts (*Figure 6A*). We found a high degree of concordance among genes expressed in vivo and in culture at early time points, but a weaker concordance at the late time points consistent with 7DIV representing a relatively early stage of in vivo development (e.g. ~P14). Nonetheless, we still saw dynamic changes in the expression of key genes marking early and late stages of cerebellar maturation in culture (*Figure 6—figure supplement 1A, B*).

We performed CUT&RUN-seq to measure changes in genome-wide distributions of H3K27me3 at 1, 3, 5, and 7 days in vitro (DIV) (*Figure 6B*, *Figure 6—figure supplement 1C*). We found that gains

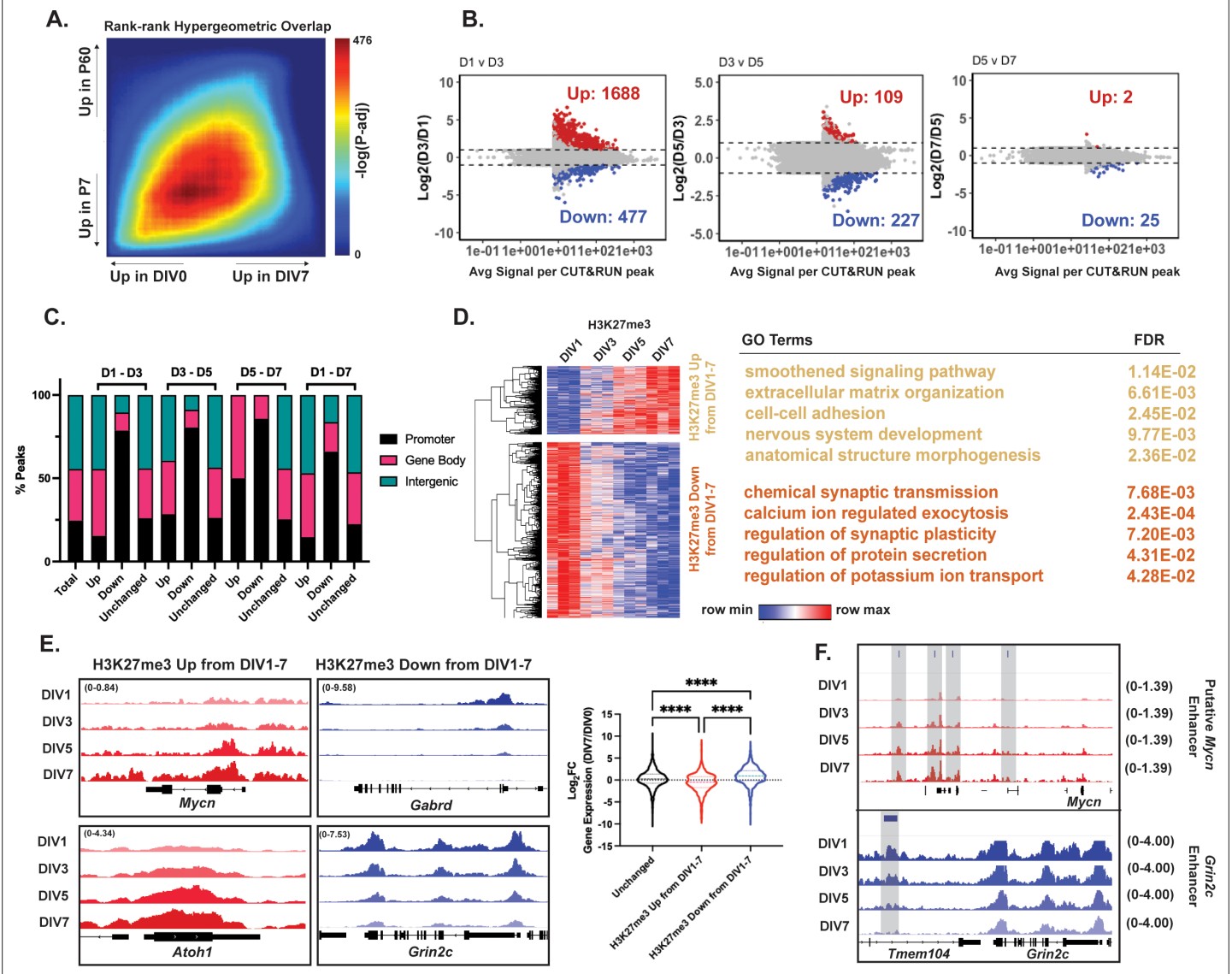

**Figure 6.** Cerebellar granule neurons (CGNs) differentiating in culture reveal temporal dynamics of histone H3 lysine 27 trimethylation (H3K27me3) changes. (**A**) Rank-rank hypergeometric overlap (RRHO) plot displaying concordance of differential gene expression measured by RNA-seq for P7-P60 cerebellum (y-axis), and cultured granule neurons for DIV0-7 (x-axis). (**B**) Differentially methylated H3K27me3 CUT&RUN peaks between DIV1-3, DIV3–5, DIV5–7, and DIV1–7. (**C**) Percentage of differential H3K27me3 CUT&RUN peaks between DIV1-3, DIV3-5, DIV5-7, and DIV1-7 annotated by genomic region (Annotation performed using ChIPseeker package, TSS +/− 3000 bp). 'Total' represents consensus H3K27me3 peaks across DIV1, DIV3, DIV5, and DIV7 combined. (**D**) (Left) Heatmap of spearman rank correlated, hierarchically clustered VST-transformed DESeq2-normalized counts of H3K27me3 CUT&RUN peaks filtered for differential promoter peaks from DIV1-7 with |Log2FC|>1 and p-adj <0.05, (Right) Corresponding Gene Ontology (GO) Terms and FDR associated with the nearest gene. (**E**) CUT&RUN tracks for H3K27me3 at example genes from 'H3K27me3 Up from DIV1-7' cluster, *Mycn* and *Atoh1*, and 'H3K27me3 Down from DIV1-7' cluster, *Gabrd,* and *Grin2c*; (right) Violin plot showing the distribution of Log2FC of gene expression measured by RNA-seq between DIV0 and DIV7 CGNs as a function of clustering performed in D (one-way ANOVA, **** indicates p<0.0001). (**F**) Tracks for H3K27me3 measured by CUT&RUN at putative postmitotic cerebellar enhancers described in *Figure 2*. Gray bars indicate putative enhancers. The numbers at the top or the right of each track indicate the y-axis scale, which is fixed for all tracks in the same set.

The online version of this article includes the following figure supplement(s) for figure 6:

**Figure supplement 1.** CUT&RUN captures genome-wide changes in histone H3 lysine 27 trimethylation (H3K27me3) in cultured cerebellar granule neurons (CGNs).

in H3K27me3 occurred quickly after cell cycle exit, with 1688 peaks significantly up between DIV1 and DIV3 CGNs, followed by 109 peaks between DIV3 and DIV5, and two peaks between DIV5 and DIV7 CGNs (*Figure 6B*). Like our observations in vivo (*Figures 1 and 2*), we saw that demethylation occurred more gradually, with 477 peaks significantly down between DIV1 and DIV3 CGNs, followed by 227 and 25 peaks between DIV3 and DIV5, and DIV5 and DIV7 CGNs, respectively. Again, consistent with our findings in vivo, we found demethylation to occur primarily at promoters (*Figure 6C*, *Figure 6—figure supplement 1D*), and gains in methylation to occur more uniformly, including at gene bodies and distal intergenic regions.

We filtered these data for differential H3K27me3 peaks between DIV1 and DIV7 at gene promoters, using a cut-off of $|Log_2FC|>1$ and p-adj <0.05, and performed hierarchical clustering. We found these peaks to cluster in two groups that gain H3K27me3 or lose H3K27me3 at DIV7 (*Figure 6D*). Genes whose promoters were in the 'H3K27me3 Up from DIV1-7' cluster were associated with GO terms indicating early CGN maturation functions such as 'smoothened signaling pathway' and 'anatomical structure morphogenesis.' Conversely, genes whose promoters were in the 'H3K27me3 Down from DIV1-7' cluster were associated with functional neuronal GO terms such as 'chemical synaptic transmission'.

We observed that genes expressed early in CGN maturation, such as *Mycn* and *Atoh1,* to gain H3K27me3 at their promoters from DIV1 to DIV7 in culture (*Figure 6E*, **left**) whereas mature synaptic genes, such as *Grin2c* and *Gabrd,* lost promoter H3K27me3 from DIV1 to DIV7 (*Figure 6E*, **center**). To determine if this relationship between H3K27me3 dynamics and developmental changes in gene expression applies broadly, we stratified our published RNA-seq data from CGNs differentiating in culture *Frank et al., 2015* by their respective promoter methylation status over development and calculated the distribution of Log2 fold change in gene expression (*Figure 6E*, **right**). We found genes in the cluster whose promoters gained H3K27me3 in culture to have a significant reduction in gene expression over time, and those that lost H3K27me3 to have a significant gain in gene expression by DIV7.

To determine whether regions of dynamic H3K27me3 in culture overlapped the same sites that were regulated in vivo, we plotted Log2 fold changes in H3K27me3 enrichment and gene expression from P7 to P60, as a function of the three clusters of H3K27me3 dynamics described in *Figure 6D*. Indeed, we found that the set of promoters that lose or gain H3K27me3 over differentiation in culture show significant developmental loss or gain, respectively, of H3K27me3 in vivo (*Figure 6—figure supplement 1E*). Our culture system also recapitulates the dynamics of H3K27me3 regulation at the enhancers we identified from our in vivo analysis as shown for the *Mycn* and *Grin2c* enhancers (*Figure 6F*). These data show that we can use CGN cultures to explore mechanisms and causality with respect to the enzymes mediating H3K27me3 turnover and gene regulation during CGN maturation.

## Disrupting H3K27me3 deposition in newly postmitotic CGNs impairs transcriptional maturation

To determine the functional contribution of H3K27me3 regulation for gene expression in maturing CGNs, we treated newly postmitotic CGN cultures at DIV1 with small molecule inhibitors of either EZH2 or KDM6A/B, called GSK-126 and GSK-J4, respectively, and measured the consequences for cell viability, histone modifications, and gene expression at DIV5. Because these drugs have not been widely used in primary neurons, we first performed a cell viability assay and measured the $IC_{50}$ values of both drugs. We determined the $IC_{50}$ values to be ~3.07 μM for GSK-J4 and ~7.32 μM for GSK-126, respectively (*Figure 7—figure supplement 1A*). Choosing concentrations below the measured $IC_{50}$ values, we first measured global levels of H3K27me3 by western blot (*Figure 7A*, *Figure 7—figure supplement 1B*), after treating with 0, 0.5, 1.0, and 2.0 μM of either drug. Treatment of CGNs with GSK-126 caused a dose-dependent loss of H3K27me3 validating the effectiveness of this drug. GSK-126 treatment also appeared to cause a global increase in H3K27ac, which is consistent with the idea that H3K27me3 and H3K27ac are mutually exclusive modifications (*Figure 7A*; *Lavarone et al., 2019*). By contrast, GSK-J4 treatment at these doses failed to cause any change in H3K27me3 detected by western or CUT&RUN (*Figure 7—figure supplement 1B–E*). Given that the $IC_{50}$ in CGNs was nearly 10-fold below the dose (25 μM) required to block the histone demethylase activity of the Kdm6s in cancer cells (*Kruidenier et al., 2012*), we concluded that the therapeutic range of GSK-J4 is too small to permit its use in neurons.

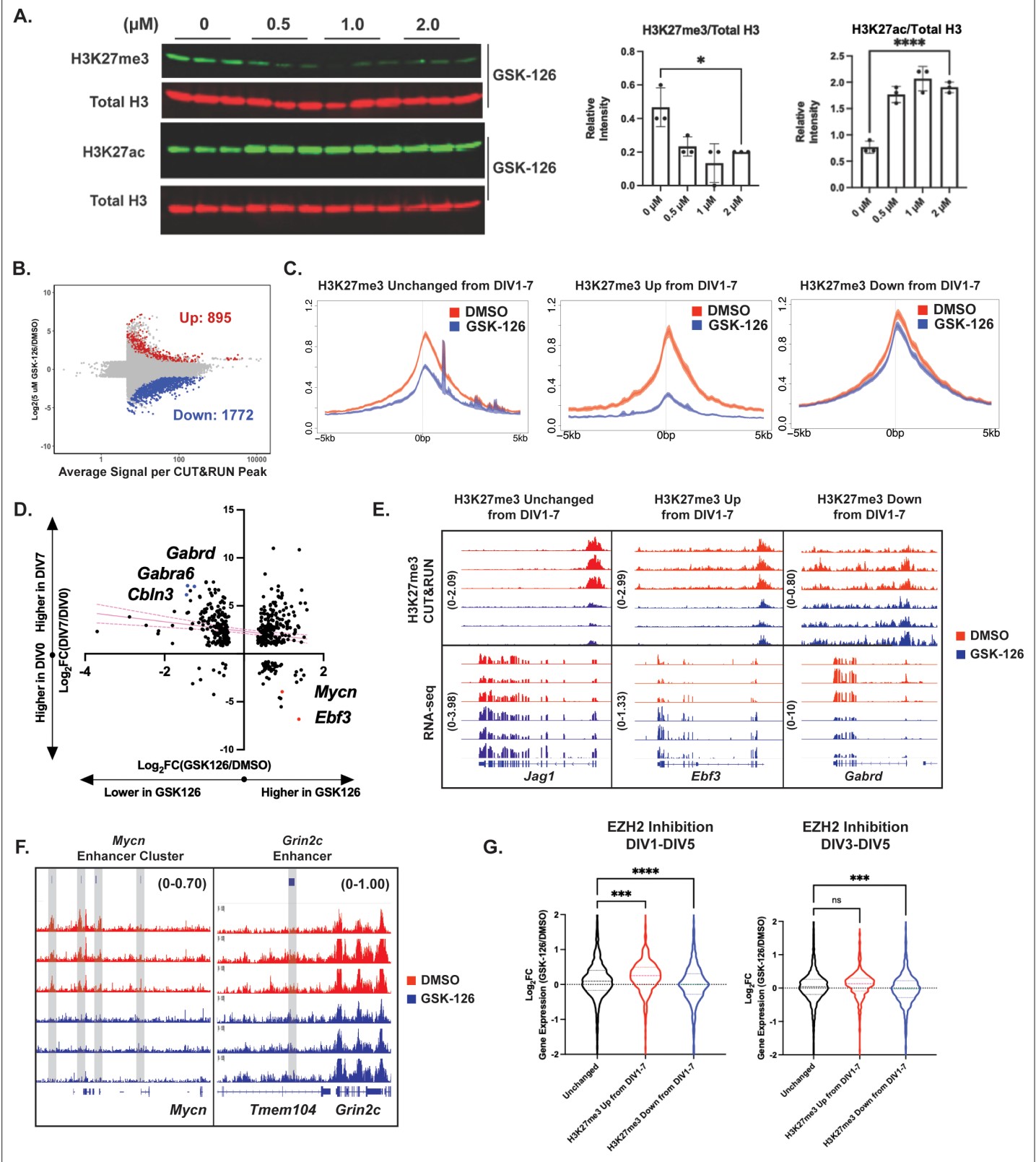

**Figure 7.** EZH2 catalytic activity temporally regulates cerebellar granule neuron (CGN) maturation by depositing histone H3 lysine 27 trimethylation (H3K27me3) at early CGN genes. (**A**) (Left) Western blot of acid-extracted histones from CGNs treated with EZH2 inhibitor GSK-126 for H3K27me3 (upper) and total Histone H3 (n=3 biological replicates) or H3K27ac (bottom) and total Histone H3 (n=3 biological replicates) (Right) Quantification of Western Blot by one-way ANOVA, *p<0.05, ****p<0.0001. (**B**) MA plot describing differential methylation due to GSK126 treatment in cultured

*Figure 7 continued on next page*

*Figure 7 continued*

CGNs (FDR <0.05 and |Log₂FC|>1) (n=3 biological replicates). (**C**) Metagene plots for H3K27me3 peaks of DMSO and 5 µM GSK-126 treated CGNs at genes within clusters (Left) All promoters with H3K27me3, (Center) H3K27me3 Up from DIV1-7, and (Right) H3K27me3 Down from DIV1-7, described in *Figure 5E*. (**D**) Relationship between gene expression changes during CGN maturation from DIV0-7 (y-axis) and due to GSK-126 treatment from DIV1-5 (x-axis). Pink lines represent the linear regression and dotted lines show the confidence interval. (**E**) Representative CUT&RUN tracks (Upper) and (Lower) RNA-seq tracks for example gene with Unchanged H3K27me3 from DIV1-7 – Jag1, gene within cluster 'H3K27me3 Up from DIV1-7' – Ebf3 and 'H3K27me3 Down from DIV1-7' - Gabrd, for DMSO and GSK126 treated CGNs. (**F**) CUT&RUN tracks at putative Mycn enhancer cluster and putative Grin2c enhancer. (**G**) Violin plot showing the distribution of Log₂FC of gene expression between DMSO and GSK-126 treated CGNs from either DIV1-5 (left) or DIV3-5 (right) as a function of clustering performed in *Figure 5D* (one-way ANOVA, **p<0.005, ***p<0.0005, ****p<0.0001).

The online version of this article includes the following source data and figure supplement(s) for figure 7:

**Source data 1.** Uncropped and annotated blot of GSK126 treated samples depicted in *Figure 7A*.

**Source data 2.** Uncropped and annotated blot of GSK126 treated samples depicted in *Figure 7A*.

**Source data 3.** Uncropped and annotated blot of GSK126 treated samples depicted in *Figure 7A*.

**Figure supplement 1.** GSK-J4 treatment at 1 µM does not affect genome wide levels of histone H3 lysine 27 trimethylation (H3K27me3), but influences the expression of late cerebellar granule neuron (CGN) genes.

**Figure supplement 1—source data 1.** Uncropped and annotated blot depicted in *Figure 7—figure supplement 1B* (GSKJ4).

**Figure supplement 1—source data 2.** Uncropped and annotated blot depicted in *Figure 7—figure supplement 1B* (GSKJ4).

**Figure supplement 2.** EZH2 inhibition redistributes histone H3 lysine 27 trimethylation (H3K27me3) across cerebellar granule neuron (CGN) genome.

To determine whether the distribution of H3K27me3 in postmitotic neurons requires the ongoing activity of EZH2, we treated CGNs on DIV1 with 5 µM GSK-126 and then ran CUT&RUN at DIV5 for H3K27me3 (*Figure 7B*, *Figure 7—figure supplement 2A*). Most of the regions that showed differential methylation upon GSK-126 treatment had reduced levels of H3K27me3 (*Figure 7B*), which is expected from the inhibition of this lysine methyltransferase. A smaller number of peaks showed increases in H3K27me3, which could arise from secondary effects of EZH2 inhibition on other processes such as EZH1 function (*Shen et al., 2008*) or histone turnover (*Chory et al., 2019*). Regions that lost H3K27me3 after GSK-126 treatment showed a similar genomic distribution to the overall distribution of H3K27me3 peaks in CGNs (*Figure 7—figure supplement 2B–C*). GSK-126-sensitive H3K27me3 promoter peaks were found at genes enriched for GO terms associated with several non-neuronal cell types, consistent with the hypothesis that PRC2 functions in postmitotic cells at least in part to maintain the chosen cell fate by repressing genes that represent alternate fates (*Ferrai et al., 2017*; *Figure 7—figure supplement 2D*).

Interestingly, a large number of H3K27me3 peaks remained unchanged after GSK-126 treatment (*Figure 7A*), suggesting that different sites of H3K27me3 are differentially sensitive to EZH2 inhibition during this period of CGN differentiation. To determine how GSK-126 sensitivity relates to differential changes in H3K27me3 during CGN differentiation, we took the three clusters of H3K27me3 peaks we derived in *Figure 6D* (H3K27me3 Up DIV1-7, Down DIV 1–7, or Unchanged) and created metagene plots to quantify the average H3K27me3 signal on DIV5 at these promoters in cells treated with the EZH2 inhibitor versus control (*Figure 7C*). Promoters that normally gain H3K27me3 over CGN differentiation were the most strongly affected by GSK-126 treatment, indicating that the activity of EZH2 in postmitotic neurons is required for the de novo acquisition of H3K27me3 peaks at these genes. Promoters where H3K27me3 is unchanged across development also showed some loss of signal, suggesting that EZH2 plays a role in the general maintenance of this repressive mark. By contrast, the set of promoters that normally lose H3K27me3 over the course of CGN development still retained substantial H3K27me3 signal at DIV5 and this was almost entirely unaffected by EZH2 inhibition. Thus, these data show that different classes of promoters are differently sensitive to the inhibition of EZH2 during postmitotic stages of differentiation. Given that we have shown the developmental regulation of H3K27me3 at these promoters to correlate with changes in gene expression, we next asked about the consequence of GSK-126 treatment for gene expression.

We performed RNA-seq on CGN cultures at DIV5 +/− GSK-126 treatment and compared the effects of EZH2 inhibition to the developmental regulation of these same genes during normal CGN differentiation (*Figure 7D*). Some of the genes that showed higher levels of expression in neurons treated with GSK-126 (e.g. *Mycn, Ebf3*) are normally downregulated as CGNs mature. These genes have EZH2-dependent developmental increases in H3K27me3 at their promoters (*Figure 7E*) and/

or enhancers (*Figure 7F*), suggesting that the developmental gain of this repressive modification facilitates turning these genes off. However, we were surprised to find that GSK-126 treatment also reduced the expression of genes such as *Gabrd* and *Grin2c* that are normally induced as CGNs differentiate, even though the levels of H3K27me3 at these gene promoters (*Figure 7E*) and enhancers (*Figure 7F*) were unaffected by GSK-126 treatment.

To determine if this relationship held broadly, we again used the gene clusters from *Figure 6D* (H3K27me3 Up DIV1-7, Down DIV 1–7, or Unchanged) and determined how gene expression changed across the genes in these clusters following EZH2 inhibition. These data show that, relative to the set of genes where H3K27me3 does not change over development, genes whose promoters normally gain H3K27me3 over development showed increased expression after GSK-126 treatment, whereas genes whose promoters normally lose H3K27me3 during development showed reduced expression (*Figure 7G*, **left**). To understand why the late genes had reduced expression after EZH2 inhibition despite having normal H3K27me3, we considered the possibility that the elevated expression of the early genes could inhibit late gene expression. To address this possibility, we performed a second experiment in which we added GSK-126 at DIV3, after H3K27me3 has already been added at many of the early-expressed gene promoters (*Figure 6D and E*). When we ran RNA-seq at DIV5 and repeated the analysis of gene expression by cluster, now we found no significant difference in the expression of the genes that gain H3K27me3 during CGN development compared with the H3K27me3 Unchanged gene cluster, but we still observed a significant reduction in expression of the late-expressed genes that lose promoter H3K27me3 over time (*Figure 7G*, **right**). These data are important because they show that proper expression of the mature CGN gene expression program requires not only the KDM6B-dependent removal of H3K27me3 at late gene promoters but also the EZH2-dependent addition of H3K27me3 elsewhere in the genome. We discuss possible mechanisms below that may explain this requirement for EZH2 in setting up the mature gene expression program for later induction.

## Discussion

In this study, we investigated the role of H3K27me3 turnover in postmitotic gene regulation, using the maturation of mouse CGNs as a model system. PRC2 activity has long been studied as a mechanism of early cell-fate specification in mammals through the deposition of H3K27me3 and silencing of fate-defining genes during the differentiation of embryonic stem cells (ESCs) into other cell types including neural progenitor cells (NPCs) and early neurogenesis (*Mohn et al., 2008*; *O'Carroll et al., 2001*; *Pereira et al., 2010*). Our study demonstrates that dynamic changes in H3K27me3 continue in postmitotic neurons at specific sites in the genome to coordinate the temporal induction of gene expression programs that underlie functional maturation. Increases in H3K27me3 occurred rapidly after cell cycle exit, whereas demethylation occurred gradually and these turnover kinetics strongly correlated with the kinetics of regulated gene expression. We identified a subset of transcriptional enhancers that show developmental regulation of H3K27me3 but observed a bias in the genomic distribution of regulated sites toward gene promoters, especially for sites of H3K27me3 demethylation. Promoters that gained H3K27me3 after cell cycle exit were enriched for genes involved in proliferation and cell-fate specification such as *Atoh1* and *Myc*. By contrast, developmentally demethylated promoters included those for genes that confer mature function on synapses such as the NMDA receptor subunit *Grin2c* and the metabotropic glutamate receptor *Grm4*. This suggested to us that there are temporally coordinated mechanisms of enzymatic H3K27me3 turnover during CGN maturation, and that postmitotic H3K27me3 demethylation may be a key step for timing functional synaptic maturation.

Our in vivo cerebellar data show that developmental loss of H3K27me3 from gene promoters is facilitated by KDM6B, and further conditional knockout of KDM6B in CGNs impaired the expression of genes that are induced as CGNs mature. *Kdm6b* expression is upregulated in newborn CGNs in the inner EGL (*Wijayatunge et al., 2018*), and we speculate that the temporal increase in expression of this enzyme may tip the balance of H3K27me3 regulation to favor histone demethylation at specific target sites. Our evidence that genes with increased H3K27me3 in the *Kdm6b*-cKO mice overlap those that show developmental loss of H3K27me3 is consistent with the possibility that KDM6B functions enzymatically to demethylate histones at these gene promoters. However, chromatin regulators including KDM6B have scaffolding functions as well as enzymatic functions (*Kim et al., 2013*; *Kim et al., 2018*; *Miller et al., 2010*; *Ohguchi et al., 2017*; *Xun et al., 2017*), thus we attempted to use pharmacological inhibition of KDM6B to validate the role of its enzymatic activity in developing CGN

gene regulation. The small molecule GSK-J4 is a relatively selective inhibitor of the KDM6 family compared with other histone demethylases, and it was previously demonstrated in primary human macrophages to block LPS-mediated histone demethylation and induction of TNFa expression with an $IC_{50}$ of 9 µM and no apparent toxicity up to 30 µM (*Kruidenier et al., 2012*). By contrast, we found 50% cytotoxicity in CGNs at a GSK-J4 concentration of just 3.1 µM and no evidence by western blot or CUT&RUN for any global or local changes in H3K27me3 when the drug was applied at the non-toxic dose of 1 µM.

Moving forward, as an alternative to pharmacological inhibition of KDM6B, genetic mutation of the JmjC domain of KDM6B offers an effective means to test its functional requirement in neuronal H3K27me3 demethylation (*Nakka et al., 2022*). Indeed, a prior study showed that in a BAC transgene, enzymatically dead KDM6B was unable to rescue perinatal death in KDM6B knockout mice, which is consistent with a requirement for the enzymatic function of KDM6B in the maturation of the respiratory circuit (*Burgold et al., 2012*). Our data now identify a set of genes induced in maturing neurons that function at synapses and require active H3K27me3 demethylation for their induction. Several of the targets of KDM6B-regulated demethylation and induction encode gene products known to play key roles in the functional maturation of synapses. These include the glutamate receptor subunit genes *Grin2c* and *Grm4,* which play important roles in cerebellar motor function and synaptic plasticity (*Kadotani et al., 1996*; *Pekhletski et al., 1996*), the GABA-A receptor delta subunit gene (*Gabrd*), which modulates the subcellular localization and neurosteroid sensitivity of GABA-A receptors (*Vicini et al., 2002*) and the neuronal pentraxin receptor gene *Nptxr,* which promotes the formation and stabilization of both excitatory and inhibitory synapses (*Lee et al., 2017*). Impaired or delayed induction of these genes in the absence of KDM6B has implications for understanding how mutations in KDM6B may lead to behavioral phenotypes in ASD via altered maturation of neural circuit function. Interestingly, mutations in another ASD-associated chromatin regulator, DNMT3A, that impair DNA methylation-dependent mechanisms of gene repression appear to be partially compensated by changes in H3K27me3 (*Li et al., 2022*). Future studies will be needed to address whether the modulation of H3K27me3 could function as a point of biological convergence downstream of multiple ASD-associated chromatin regulators.

Our observation that H3K27me3 removal occurs at only a subset of genes that are developmentally induced during CGN maturation raises the question of how these promoters are marked for methylation and demethylation. Through an unbiased approach, we showed that promoters that lose H3K27me3 during CGN maturation are strongly enriched for binding of the transcription factors ZIC1/2; thus, one possibility is that an interaction between KDM6B and the ZICs selects these genes for induction in maturing neurons. ZIC1/2 are strong drivers of CGN maturation (*Frank et al., 2015*), and mutation of these genes is associated with cerebellar disorders, showing their importance in CGN development (*Blank et al., 2011*; *Twigg et al., 2015*). ZIC1 and 2 are zinc-finger binding proteins that exhibit dynamic DNA binding properties during cerebellar maturation, and they regulate the expression of both early and late CGN maturation genes (*Frank et al., 2015*). Given the strong association of H3K27ac enrichment at these promoters, however, it is likely that ZIC1/2 binding is a consequence and not a cause of H3K27-demethylation at promoters. It would, therefore, be essential to measure the genome-wide enrichment patterns of KDM6B during CGN maturation, in order to identify its binding partners and direct targets. Existing literature investigating KDM6B dynamics across the epigenome is sparse owing to the lack of well-validated, commercially available KDM6B antibodies. Nevertheless, KDM6B has been shown to localize to promoters and enhancers of neurogenic genes in NSCs (*Estarás et al., 2012*; *Park et al., 2014*), and more recently to induce muscle regeneration gene *Has2*, by removing H3K27me3 from its promoter, to initiate muscle repair in response to injury (*Nakka et al., 2022*). KDM6B targeting to these specific loci might be facilitated through interactions with TFs (*Dai et al., 2010*; *Estarás et al., 2012*), and chromatin remodeling complexes (*De Santa et al., 2007*). For example, KDM6B has been shown to strongly associate with the ATP-dependent chromatin remodelers mSWI/SNF or BAF complex, by interacting with its subunits which in turn potentiate its activity (*Narayanan et al., 2015*). This interaction has been shown to play a role in neuronal development (*Narayanan et al., 2015*; *Nguyen et al., 2018*).

Interestingly, we find that maturation of CGN transcriptional programs not only depends on the loss of H3K27me3 at genes that turn on late, but also requires the addition of H3K27me3 at regions that gain this mark in newly postmitotic neurons. Using CUT&RUN in cultured CGNs, we discovered that

the rapid increases in H3K27me3 that accumulate in newly postmitotic neurons are highly sensitive to inhibition of EZH2 with GSK-126. Failure to gain H3K27me3 at these sites in differentiating neurons not only prevented the developmental downregulation of CGN progenitor markers like *Mycn, Jag1,* and *Ebf3* but also impaired the developmental induction of late genes like *Gabrd*, *Grm4,* and *Grin2c*. The impact of GSK-126 on the mature CGN transcriptional program, however, was not due to changes in H3K27me3 regulation at promoters of the late genes themselves. This shows that the postmitotic addition of H3K27me3 at specific genomic sites is crucial in regulating the induction of genes that turn on well after neuronal cell fate is determined.

These data raise the question of how the PRC2 complex function might indirectly regulate the activation of late gene expression programs over a period of days. One prior study that reported the consequences of knocking out EZH2 in postmitotic neuronal cultures showed that this resulted in the loss of H3K27me3 at the promoter of the *Prdm13* gene and enhanced expression of this transcriptional repressor, which they suggested then led indirectly to reduced expression of its target genes in the knockout neurons (*Buontempo et al., 2022*). Though a similar mechanism could be at play in our cultures, *Prdm13* expression and promoter H3K27me3 were unaffected by EZH2 inhibition in our DIV5 CGNs, thus we have considered alternative explanations. Notably, developmental gains in H3K27me3 during postmitotic differentiation are relatively equally split between promoters, gene bodies, and distal intergenic regions (*Figure 1D*). By comparing these sites of H3K27me3 regulation to other chromatin marks, we found that the majority of non-promoter peaks of H3K27me3 do not overlap enhancers. These data raise the possibility that rather than functioning to control transcriptional regulatory elements directly, H3K27me3 might contribute to gene regulation through an alternate mechanisms such as by impacting chromatin architecture. PRC2-dependent deposition of H3K27me3 has been shown in some cases to contribute to physical interactions between enhancers and their target genes by anchoring chromatin loops (*Cruz-Molina et al., 2017*; *Kraft et al., 2022*). These loops may then poise genes for enhancer activation later upon receiving a developmental stimulus. The gain of H3K27me3 in early differentiating CGNs could serve a similar function to establish such enhancer contacts with the promoters of late genes such that when KDM6B demethylates these promoters, the enhancer can permit activation. In the future, investigating dynamics in higher-order chromatin structure through low-input techniques such as HiCAR (*Wei et al., 2022*) might help inform how H3K27me3 dynamics contribute to gene regulation in postmitotic neurons.

# Materials and methods
## Animal husbandry
We performed all procedures under an approved protocol from the Duke University Institutional Animal Care and Use Committee. *Kdm6b* floxed mice described in *Manna et al., 2015* were obtained from Jackson Labs (Stock #029615, RRID #IMSR_JAX:029615). These mice contain *loxP* sites flanking exon 14–20, encoding the catalytic Jumonji-C domain, of the mouse *Kdm6b* gene on chromosome 11. *Atoh1-Cre* mice were obtained from Jackson Labs (Stock #011104, RRID #IMSR_JAX:011104). Mice were genotyped by ear clipping at the time of weaning using protocols described on the Jackson Labs website. CD-1 IGS mice (#Strain 022, #IMSR_CRL:022) and C57BL/6NCrl mice (#Strain 027, RRID #IMSR_CRL:027) were obtained from Charles River Laboratories. Both male and female mice were used for all experiments in this study.

## CGN culture
Cerebella of CD1-IGS mice were dissected at P6-P8. Cells were dissociated and run on a Percoll gradient to isolate cerebellar GNPs as previously described (*Bilimoria and Bonni, 2008*). GNPs were plated at a density of 1 million cells/well in a 24-well plate for RNA or protein isolation. At a day in vitro 2 (DIV2) CGNs were treated with 1 μM Cytosine Arabinoside (AraC) (Sigma #C1768) to block the division of any non-neuronal cells. The KDM6 family inhibitor GSK-J4 (Sigma #SML0701) (*Kruidenier et al., 2012*) and the EZH2-specific inhibitor GSK-126 (Cayman #15415) (*McCabe et al., 2012*) were first used in a dose-response assay to determine $IC_{50}$ in CGNs. For this, Cell Titer Glo (Promega #G7570) was performed on CGNs cultured in a 96-well plate at a density of 100,000 cells/well, with inhibitors added one time at DIV1 and CGNs processed for downstream assays at DIV5. Further

experiments with inhibitors were performed by treating CGNs one time at either DIV1 or DIV3 and processing for downstream assays at DIV5, using doses as specified in the text.

## RT-qPCR

CGNs were lysed in TRIzol Reagent (Thermo #15596026) at a volume of 0.25 mL per 1 million cells for RNA purification. Isolated RNA was analyzed using a nanodrop to determine concentration and purity, after which it was treated with DNase I (NEB #M0303S). We made Oligo(dT) primed cDNA using Superscript-II Reverse Transcriptase (Invitrogen #19064014). qPCR was carried out on an Applied Biosystems Quantstudio 3 Thermal cycler using Power SYBR Green PCR Master Mix (Thermo #4367659) and PCR primers targeting the gene/locus of interest (*Supplementary file 1*). qPCR data was processed and analyzed using the $2^{-ddC(t)}$ method (*Livak and Schmittgen, 2001*).

## RNA-seq

RNA for RNA-seq was obtained by processing cerebellar tissue or CGN cultures with Trizol reagent followed by clean up on the Zymo Direct-zol RNA miniprep kit (Zymo #R2052). RNA purity was measured to ensure samples had A260/280 and A260/230 values >1.9. RNA was then polyA enriched and 150 bp paired-end sequencing was performed by Novogene, Inc on an Illumina Hi-seq 2000 machine. Unless specifically noted in the text, three independent biological replicates were sequenced for each condition.

## Chromatin immunoprecipitation

### Cerebellar tissue

We pooled cerebellum from three C57BL/6NCrl (Charles River Laboratories IMSR Cat# CRL:027, RRID: IMSR_CRL:027) P7 mice, two P14 mice and one P60 mouse for each biological replicate. Both male and female mice were used. Cerebellum samples were dounced in a glass homogenizer containing 2 ml of 1% formaldehyde (w/v) PBS buffer per sample and kept at 25 °C for 15 min, washed twice with cold PBS, lysed in 600 µl lysis buffer (1% SDS (w/v), 10 mM EDTA, and 50 mM Tris, pH 8.1). The cross-linked material was sonicated with a Bioruptor (Diagenode), with output set to 'high' with 30 s on/off cycles to an average size range of 150–350 bp which was then determined by agarose gel electrophoresis. Sonicated chromatin was centrifuged for 10 min at 14,000 RPM at 4 °C, after which the supernatant was diluted 10-fold in dilution buffer (0.01% SDS, 1.1% Triton X-100, 1.2 mM EDTA, 16.7 mM Tris-HCl, pH 8.1, 167 mM NaCl). Prior to immunoprecipitation, 6 µl of antibody (anti-H3K27Me3, Active Motif 39155, RRID: AB_2561020) was incubated with 100 µl of Dynabeads Protein G (Invitrogen 10004D) in PBS for 4–6 hr at 4 °C. The antibody-bead conjugate was then washed twice with cold PBS, and added to 6 ml of cell lysis for overnight immunoprecipitation at 4 °C. The following day, bead-bound DNA-protein complexes were washed, eluted, and purified using a PCR purification kit (Cat# 28104, Qiagen). ChIP-seq libraries were made using the MicroPlex Library Preparation kit V2 (C05010012, Diagenode); and 50 bp single-end sequencing was performed at the Duke Sequencing and Analysis Core Resource on a Hi-Seq 2000 machine. Unless specifically noted in the text, three independent biological replicates were performed for each developmental time point, and ChIP reagents were prepared according to the Millipore 17–295 ChIP kit.

## Nuclear isolation of CGNs for CUT&RUN

CGNs were scraped into 1 X DPBS at 0.25 mL per 1 million cells and then 'pop-spun' according to the REAP method (*Suzuki et al., 2010*), until the rotor reached 7000 rpm. Pellets were then washed once with 1 X DPBS and pop-spun again, after which they were resuspended in Nuclei Isolation Buffer (20 mM HEPES pH 7.9, 10 mM KCl, 2 mM Spermidine, 0.1% v/v Triton X-100, 20% v/v glycerol), incubated on ice for 5 min and then spun at 2000 g for 5 min at 4 °C. After this step the supernatant was removed, and pelleted nuclei were then resuspended in Nuclei Storage Buffer (20 mM Tris-HCl pH 8.0, 75 mM NaCl, 0.5 mM EDTA, 50% v/v glycerol, 1 mM DTT, 0.1 mM PMSF) and stored in –80 °C until ready to process.

## CUT&RUN

CUT&RUN was performed on nuclei isolated from CGN cultures using the CUTANA ChIC/CUT&RUN kit (EpiCypher #14–1408) as per manufacturer guidelines. Specific changes made to the protocol

are noted here. Nuclei in Nuclei Storage Buffer were pelleted and resuspended in Nuclei Isolation Buffer. Nuclei were then incubated with activated ConA beads. Antibodies used for CUT&RUN were: H3K27me3 (Active Motif 39155, RRID: AB_2561020), H3K4me3, and IgG (positive and negative controls included in the kit). CUT&RUN libraries were made using the NEB Ultra II DNA Library Prep Kit for Illumina (NEB #E7645L), and NEBNext Multiplex Oligos for Illumina (96 Unique Dual Index Primer Pairs) (NEB #E6440S). Library cleanup was performed prior to and after PCR amplification using 0.8 X Kapa Hyperpure beads (Roche #08963851001). PCR amplification was performed with the following parameters as described in the EpiCypher CUT&RUN kit: (1) 98 °C, 45 s; (2) 98 °C, 15 s; 60 °C, 10 s × 14 cycles; (3) 72 °C, 60 s. Libraries were then pooled and 50 bp paired-end sequencing was performed at the Duke Sequencing and Analysis Core Resource on a NovaSeq 6000 S-Prime flow cell. Three independent biological replicates were performed for each developmental time point and drug treatment.

## Sequencing data processing
### ChIP-seq
ChIP-seq reads were quality scored and screened through fastQC, trimmed for adapter sequences using Trimmomatic 0.38, and then aligned to the mouse GRCm38.p4 reference genome using STAR 2.7.2b. Alignments were filtered for mapped reads using samtools and then normalized using the bamCoverage function of the deepTools2.0 suite. ChIP-seq tracks were generated using bamCoverage, to generate continuous BigWig files. Peak calling was performed using Macs2, with parameters for broad peaks and an FDR threshold of 0.01. H3K4me1 ChIP-seq data, obtained from *Ramirez et al., 2022* was additionally processed by normalizing IP'd read to input using the bamCompare function of the deepTools2.0 suite due to their lower signal-to-noise ratio compared to H3K27me3, H3K4me3, and H3K27ac data.

### CUT&RUN
CUT&RUN reads were processed the same as ChIP-seq reads until peak-calling. Peak calling was performed using Macs2, with parameters for broad peaks and an FDR threshold of 0.1.

### RNA-seq
RNA-seq reads were processed similarly to ChIP-seq reads until and excluding peak calling. Read counts for genes were generated using HTseq2 count, with the feature type set to 'exon,' and the Gencode GRCm38.p4 GTF annotation file.

## Western blot
### Cerebellar tissue
For the preparation of protein lysates from cerebellar tissue, we pooled cerebellum from two P7 mice, one P14, and one P60 mouse for each biological replicate. Cerebellum samples were dounced 50 times with PBS in a glass homogenizer, centrifuged at 8000 × g for 5 min, and washed twice with cold PBS, then pellets were re-suspended in 300 µL extraction buffer (10 mM HEPES, pH 7.9, 1.5 mM MgCl$_2$, 10 mM KCl), supplemented with 1 X protease inhibitor cocktail (Roche, Indianapolis, IN). Thereafter, 60 µL of 1 N HCl was added to the cell lysates and kept on ice for 30 min, following which the cell lysates were centrifuged at 11,000 × g for 10 min at 4 °C. Protein concentration of acid-soluble supernatants was measured using a Pierce BCA protein assay kit (Cat # 23227, Thermo). 20 µg of acid-soluble supernatant was used for western blots. Samples were loaded onto wells of a 10–20% gradient gel and run at 150 V for 30–40 min in running buffer (25 mM Tris-HCl, 192 mM Glycine, 0.1% w/v SDS) after which protein was transferred onto a PVDF membrane (BioRad #1620177) in transfer buffer (25 mM Tris-HCl, 192 mM Glycine, 20% v/v Methanol) for 1 hr at 100 V or overnight at 30 V. The membrane was blocked for 1 hr in 5% (w/v) BSA (Sigma #A3059) in TBST. After the blocking step, the membrane was incubated with primary antibody overnight at 4 °C. The following day, the primary antibody was removed, after which the membranes were washed three times with TBST for 5 min each, incubated with secondary antibody for 1 hr, and then again washed three times with TBST for 5 min each. At the end of this, membranes were imaged using the Li-Cor Odyssey using appropriate exposure times for the target antibodies.

## CGN culture

Cells were scraped into PBS from six-well plates at the relevant endpoints, centrifuged at 200 × g for 5 min, and washed twice with PBS. Pellets were then resuspended in 100 µL of extraction buffer (described above), after which 20 µL of 1 N HCl was added. Samples were hereafter processed as described above.

## Antibodies

The antibodies used for western were H3K27Ac (Abcam ab4729), H3K27Me3 (Active Motif 39155, RRID: AB_2561020), and Total H3 (Tissue: Millipore 05–499, Culture: Cell Signaling 9715). Western blot signals were detected by Li-Cor Odyssey InfraRed Imaging System (Lincoln, NE) with the secondary antibodies, CF770 goat anti-Rabbit, (Biotium 20078), and CF 680 goat anti-mouse, (Biotium 20065). Protein bands were quantified using ImageJ.

## Statistics and reproducibility

Biological replicates (in vivo) are defined as individual mouse cerebellar tissue (per replicate) or (*in culture*) CGNs derived from pooled cerebellar tissue from mouse litters (per replicate). Statistics on Western blot quantification were performed using ordinary one-way ANOVA followed by Tukey's multiple comparisons test. The R package DESeq2 (*Love et al., 2014*) was used for read-count normalization and differential binding analysis. R package ChIPseeker (*Yu et al., 2015*) was used to annotate ChIP-seq peaks using a window of TSS ± 3000. 'Promoters' is a group of ChIPseeker terms 'Promoter ( ≤ 1 kb), Promoter (1–2 kb), and Promoter (2–3 kb). Gene Body is a group of terms including 5' UTR, 3' UTR, Intron, and Exon. Statistics for violin plots comparing distributions of Log2 fold changes of DESeq2 normalized read-counts for RNA-seq, ChIP-seq, and DNAse-seq data between clusters of genes, were performed using ordinary one-way ANOVA followed by Tukey's multiple comparisons test unless not normally distributed, in which case a Kruskal-Wallis test was performed. Heatmaps were generated using the Broad Institute's Morpheus tool: https://software. broadinstitute.org/Morpheus, with the following clustering parameters: hierarchical clustering, complete linkage, one minus spearman rank correlation, and cluster by rows. VST-transformed counts used to generate heatmaps and clusters are provided in *Supplementary file 2*. Gene ontology analysis was performed using the GO consortium GO tool (*Ashburner et al., 2000*) using terms related to 'Panther GO biological process slim' (*Mi et al., 2019*). Genes used to generate GO terms are provided in *Supplementary file 3*. Metagene plots were created using SeqPlots (*Stempor and Ahringer, 2016*), by inputting bigwig and BED files (peak coordinates for plots centered around peaks, and gene coordinates for plots centered around genes) and setting a window of peak center/ TSS +/- 5000 bp.

## GEO accession codes

H3K27me3 ChIP-seq data for P7, P14, and P60 cerebellum, WT and *Kdm6b*-cKO cerebellum; RNA-seq data for WT and *Kdm6b*-cKO cerebellum; CUT&RUN and RNA-seq data for cultured CGNs can be accessed at GEO: GSE212441. RNA-seq data for P7, P14, and P60 cerebellum and DIV0 and DIV7 CGNs, H3K27ac, ZIC1/2 ChIP-seq data, and DNAse-seq data for P7 and P60 cerebellum were adapted from *Frank et al., 2015* and can be accessed at GEO: GSE60731. H3K4me3 ChIP-seq data from P6 and P22 cerebellum were obtained and adapted from *Yamada et al., 2014* and can be accessed at GEO: GSE57758. H3K4me1 ChIP-seq data from the P9 cerebellum was obtained and adapted from *Ramirez et al., 2022* and can be accessed at GEO: GSE183697. H3K4me3 PLAC-seq tracks were obtained from *Yamada et al., 2019* and can be accessed at GEO: GSE127995.

## Acknowledgements

Figure schematics in *Figures 1A and 4A* were created using biorender.com. This work was supported by NIH grant R01NS0988804. We thank Yue Yang at Northwestern University for providing us with the H3K4me3 PLAC-seq loop data from the P56 cerebellum. This work was previously uploaded to BioRxiv at https://doi.org/10.1101/2022.10.10.511582.

## Additional information

### Competing interests
Anne E West: Reviewing editor, *eLife*. The other authors declare that no competing interests exist.

### Funding

| Funder | Grant reference number | Author |
|---|---|---|
| National Institutes of Health | R01NS0988804 | Anne E West |

The funders had no role in study design, data collection and interpretation, or the decision to submit the work for publication.

### Author contributions
Vijyendra Ramesh, Conceptualization, Data curation, Formal analysis, Validation, Investigation, Visualization, Methodology, Writing - original draft, Writing - review and editing; Fang Liu, Conceptualization, Investigation, Methodology; Melyssa S Minto, Formal analysis; Urann Chan, Methodology; Anne E West, Conceptualization, Resources, Supervision, Funding acquisition, Investigation, Methodology, Writing - original draft, Writing - review and editing

### Author ORCIDs
Vijyendra Ramesh ⓘ http://orcid.org/0000-0001-8818-6863
Melyssa S Minto ⓘ http://orcid.org/0000-0002-5438-7285
Anne E West ⓘ http://orcid.org/0000-0003-0846-139X

### Ethics
We performed all procedures under an approved protocol from the Duke University Institutional Animal Care and Use Committee, (#A035-20-02).

### Decision letter and Author response
Decision letter https://doi.org/10.7554/eLife.86273.sa1
Author response https://doi.org/10.7554/eLife.86273.sa2

## Additional files

### Supplementary files
• MDAR checklist
• Supplementary file 1. RT-qPCR primers.
• Supplementary file 2. Input DESeq2 normalized counts used for heatmaps.
• Supplementary file 3. Input gene lists for GO analysis.

### Data availability
H3K27me3 ChIP-seq data for P7, P14 and P60 cerebellum, WT and Kdm6b-cKO cerebellum; RNA-seq data for WT and Kdm6b-cKO cerebellum; CUT&RUN and RNA-seq data for cultured CGNs can be accessed at GEO: GSE212441. RNA-seq data for P7, P14 and P60 cerebellum and DIV0 and DIV7 CGNs, H3K27ac, ZIC1/2 ChIP-seq data and DHS-seq data for P7 and P60 cerebellum were adapted from *Frank et al., 2015*. H3K4me3 ChIP-seq data from P6 and P22 cerebellum were obtained and adapted from *Yamada et al., 2014* and can be accessed at GEO: GSE57758. H3K4me1 ChIP-seq data from P9 cerebellum was obtained and adapted from *Ramirez et al., 2022* and can be accessed at GEO: GSE183697. H3K4me3 PLAC-seq tracks were obtained from *Yamada et al., 2019* and can be accessed at GEO: GSE127995. Source data files for western blots in Figure 1, 7, Figure 1—figure supplement 1 and Figure 7—figure supplement 1 are included. Primers used to perform RT-qPCR have been provided in Supplementary file 1. DESeq2-normalized counts used to generate heatmaps in the study have been provided in Supplementary file 2. Gene lists used for Gene Ontology analyses have been provided in Supplementary file 3. Processed data, DESeq2 outputs and more are provided at https://github.com/WestLabDuke/CGN_H3K27me3_Dynamics, (copy archived at *West Lab at Duke, 2023*).

The following dataset was generated:

| Author(s) | Year | Dataset title | Dataset URL | Database and Identifier |
|---|---|---|---|---|
| Ramesh V, Liu F, Minto M, Chan U, West AE | 2022 | Bidirectional changes in postmitotic H3K27me3 distributions underlie cerebellar granule neuron maturation dynamics | https://www.ncbi.nlm.nih.gov/geo/query/acc.cgi?acc=GSE212441 | NCBI Gene Expression Omnibus, GSE212441 |

The following previously published datasets were used:

| Author(s) | Year | Dataset title | Dataset URL | Database and Identifier |
|---|---|---|---|---|
| Frank CL, Liu F, Wijayatunge R, Song L, Biegler MT, Yang MG, Vockley CM, Safi A, Gersbach CA, Crawford GE, West AE | 2015 | Regulation of chromatin accessibility and Zic binding at enhancers in the developing cerebellum | https://www.ncbi.nlm.nih.gov/geo/query/acc.cgi?acc=GSE60731 | NCBI Gene Expression Omnibus, GSE60731 |
| Yamada T, Yang Y, Hemberg M, Yoshida T, Cho HY, Murphy JP, Fioravante D, Regehr WG, Gygi SP, Georgopoulos K, Bonni A | 2014 | Promoter Decommissioning by the NuRD Chromatin Remodeling Complex Triggers Synaptic Connectivity in the Mammalian Brain | https://www.ncbi.nlm.nih.gov/geo/query/acc.cgi?acc=GSE57758 | NCBI Gene Expression Omnibus, GSE57758 |
| Ramirez M, Goldowitz D | 2021 | Temporal analysis of enhancers during mouse brain development reveals dynamic regulatory function and identifies novel regulators of cerebellar development | https://www.ncbi.nlm.nih.gov/geo/query/acc.cgi?acc=GSE183697 | NCBI Gene Expression Omnibus, GSE183697 |
| Yamada T, Yang Y, Valnegri P, Juric I, Abnousi A, Markwalter KH, Guthrie AN, Godec A, Oldenborg A, Hu M, Holy TE, Bonni A | 2019 | Sensory experience remodels genome architecture in neural circuit to drive motor learning | https://www.ncbi.nlm.nih.gov/geo/query/acc.cgi?acc=GSE127995 | NCBI Gene Expression Omnibus, GSE127995 |

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
