## [Editor Report]

The significance of the work is the choice of analyzing the coordination of chromatin modifications guiding cerebellar granule maturation, an important biological process that is uniquely suited for such an analysis. A strength of the work is their comprehensive findings revealing changes in specific chromatin modifications in promoters and additional regulatory elements that come into play during sequential stages of maturation. Authors also provide evidence for gene expression changes that correlate with the changes in chromatin status, using tools available in vivo and in cell culture. The paper was accepted because the authors satisfactorily addressed the concerns of both reviewers.

---

## [Decision Letter]

**Decision letter after peer review:**

Thank you for submitting your article "Bidirectional regulation of postmitotic H3K27me3 distributions underlie cerebellar granule neuron maturation dynamics" for consideration by *eLife*. Your article has been reviewed by 2 peer reviewers, one of whom is a member of our Board of Reviewing Editors, and the evaluation has been overseen by Kevin Struhl as the Senior Editor. The reviewers have opted to remain anonymous.

Essential revisions:

1) Authors need to expand the narrow focus of the work related to promoter analysis and extend it to include enhancers, using the most recent databases, such as the database for cerebellar enhancers noted in the reviews.

2) Authors need to provide data in the cultures, or in vivo if relevant databases exist, for the presence/absence of the H3K4me3 modification on early and late promoters, as well as the H3K4me1mark for enhancers. They should also explain/speculate in the discussion why they think there is no phenotype in the KO mice with respect to cerebellar development, despite the findings on changes in the early/late genes? If they have a more detailed time course of changes in late gene expression, when early genes are repressed persistently after treatment with EZH2 inhibitor, they should add this information, otherwise discuss any relevant information from previous studies bearing on this issue.

3). We are not requesting new experiments related to autism, but if they choose to emphasize this point, authors are requested to discuss how their results with PRC2 provide insight into the epigenetic basis for autism, in the context of previously published studies by others on additional chromatin factors, as mentioned in the reviews.

*Reviewer #1 (Recommendations for the authors):*

In this work, authors exploit the prolonged in vivo maturation of postmitotic cerebellar granule neurons to test how changes in the chromatin modification, H3K27me3, may coordinate an orderly change in RNA expression profiles during maturation. The authors show that normal in vivo cerebellar development is associated with temporal gains and losses of H3K27me3 at promoters. In a conditional KO mouse for the lysine demethylase, KDM6B, late-expressing CGN genes, encoding mature neuronal functions, are abnormally hypermethylated and repressed. Some of these genes, which lose H3K27me3 during normal development, and are expressed, are shown to be associated with ZIC family transcriptional activators. Conversely, authors treat CGN primary cultures with an inhibitor of the PRC2 catalytic subunit, the lysine methyltransferase, EZH2. As expected, some genes lose H3K27me3 at their promoters with treatment. These genes are found to encode "early" non-neuronal/developmental processes, are repressed during normal maturation, and are up-regulated after treatment compared to controls. Unexpectedly, while there is no change in H3K27me3 methylation of "late" genes, authors observe a decrease in their expression levels with inhibitor treatment. Based on their findings, authors propose a bidirectional regulation by H3K27me3 which coordinates gene expression programs underlying functional neuronal maturation. Because mutations in KDM6B are a risk factor for autism, authors also propose that the work may provide insight into this disorder.

Exploiting the prolonged maturation in CGN neurons is an elegant way to dissect the regulation of early and late gene expression events over developmental transitions. However, the advance here is somewhat incremental in light of two previous studies by this lab showing effects of CGN maturation and polycomb function in vivo, and in culture, using the same KDM6B animal model (Frank et al., 2015; Wijayatunge et al., 2018). Innovation is also impacted by a wealth of information that exists already for roles for H3K27me3 in both cell fate decisions and maturation, in the nervous system and elsewhere, including a description of waves of H3K27me3 in oligodendrocyte lineage (Bartosovic and Castelo-Branco, 2022). In this work by authors, both early neuronal and non-neuronal genes are also affected. While some of the relevant previous references are cited, it is not in the context of how authors' new work adds to the biological significance of what was already known, or how the current results dramatically alter thinking about the functions of the PRC2 complex or H3K27me3 generally. The bidirectional effect here is interesting but remains unexplored.

There is also concern about the narrow focus of the work, which is primarily bioinformatics driven. The conclusion in the abstract that they have uncovered gene programs underlying functional neuronal maturation is a bit overstated because there are no functional studies in this work. A previous study by authors cited in the paper indicated that the conditional KO mouse model does not show any effects on the overall development of the cerebellum, and no phenotypes are provided in the current study after the manipulation of the enzymes. Therefore, the biological significance of bidirectionality is not known. The focus of the current work would be broadened by deeper analysis related to the regulation of early and late enhancers. The field of enhancer regulation in the nervous system is getting quite mature, including a resource of dynamic changes in cerebellum enhancers during development (Ramierez et al., 2022). New experiments, and/or a deeper analysis of chromatin events, would increase interest in a general audience. Motif analysis of enhancers could provide clues as to whether other protein factors are recruited in conjunction with H3K27me3 to early genes. The authors use CUT and RUN in the culture system for H3K27me3, but this approach can be used with other histone modification antibodies. The work could be broadened to investigate, for example, whether the promoters are marked by both H3K4me3 and H3K27me3. Is the chromatin of slow and fast maturation genes in culture differentially marked? This could also be done in vivo.

While the cerebellum is a region associated with autism, DNMT3a is also linked to autism, several BAF subunits are mutated in autism, and new enhancers in the cerebellum have been linked to autism recently (Ramierez et al., 2022). Without knowing whether there is a convergence of these other pathways on early or mature genes marked, or not, with H3K27Me3, it is not clear how the present work advances understanding of this complex disorder.

*Reviewer #2 (Recommendations for the authors):*

This study provides numerous well-controlled experiments to define the dynamics of H3K27me3 in cerebellum development. I have some additional comments on areas that could be addressed:

– The analysis of the effects of KDM6B deletion on H3K27me3 and gene expression makes a convincing case that the activity of this enzyme is important for the removal of H3K27me3 and gene activation during postnatal maturation, but a direct assessment of the overlaps of the genes that are differentially expressed in development and those that are differentially expressed upon KDM6B deletion is not prominently presented. The degree to which there is overlap between these gene lists, and the identity of these genes could be helpful for understanding the functional importance of KDM6B and H3K27me3 removal during neuronal maturation.

– Although the analysis of H3K27me3 dynamics at promoters and gene bodies is informative, the authors do not provide any analysis of enhancers in the study. The study would be significantly strengthened if some assessment of H3K27me3 was carried out at these sites. Given that the authors have access to histone modification profiles that would allow them to identify putative enhancers in the cerebellum (H3K27ac Frank et al. 2015), it should be possible for the authors to carry out this analysis without completing additional experiments.

– The study takes advantage of the CGN culture system in order to do insightful pharmacology experiments. These cells show similar H3K27me3 dynamics as in vivo samples, making it a suitable system for their analyses. However, the authors do not make a direct comparison of the genes that have changes in H3K27me3 and gene expression in this system and the genes that show similar changes in vivo. A direct comparison of the genesets by overlap analysis would be helpful for the reader to know if the same genes are affected in both systems, or if the concept of dynamic H3K27me3 is the same, but the specific loci affected are context dependent.

– It would be helpful if the authors could better describe how the clusters in Figure 2A were defined. For example, the dendrogram could be included and show the remaining genes, not in the clusters presented. For many "fast" and "slow" peaks the profiles look quite similar. It appears that differences between these clusters in terms of associated gene expression might be more distinct if the slow population was more restricted to the genes at the bottom of that cluster that displays much more clear persistence of methylation through P14.

– The quantitative analysis of DHS in Figure 4E (left) shows a reduction in DHS signal at P7 to P60 "H3K27me3 up" promoters but the aggregate plot (right) appears to show an increase in DHS signal at these sites. No mention of these contradictory results is made in the text. Do the authors have an explanation for this disparity?

– The gene ontology analysis that is performed in the study does a convincing job of showing the general classes of molecules affected by H3K27me3 dynamics, but very limited analysis or discussion is made of specific genes that are dysregulated and how they may functionally influence CGN maturation. While the characterization of the functional effects of disrupted expression for these genes may be beyond the scope of this manuscript, the study would be improved if a few examples were noted and discussed.

---

## [Author Response]

Essential revisions:1) Authors need to expand the narrow focus of the work related to promoter analysis and extend it to include enhancers, using the most recent databases, such as the database for cerebellar enhancers noted in the reviews.

We truly thank the reviewers for the suggestion to assess developmental changes in H3K27me3 at enhancers as marked by non-promoter peaks of H3K4me1 and H3K27ac. We feel these analyses have substantially improved the study. We have added these data to a new figure, which is now Figure 3 in the revised manuscript. We have also added data to what is now Figure 3 – supplement 1. These data are presented in the Results section of the manuscript on Page 10.

These data show that changes in H3K27me3 can indeed be correlated with developmental changes in the activity of a small number (hundreds) of cerebellar enhancers. However, they also show that most non-promoter sites of differential H3K27me3 signal do not overlap chromatin marks of enhancers and most developmentally regulated cerebellar enhancers do not overlap sites of differential H3K27me3 regulation.

We conclude in the discussion on Pg. 23, “We identified a subset of transcriptional enhancers that show developmental regulation of H3K27me3 but observed a bias in the genomic distribution of regulated sites toward gene promoters, especially for sites of H3K27me3 demethylation.”

These data also help to deepen our model of the requirement for EZH2 in the expression of CGN maturation genes. On Pg. 27 we say, “Notably, developmental gains in H3K27me3 during postmitotic differentiation are relatively equally split between promoters, gene bodies, and distal intergenic regions (Figure 1D). By comparing these sites of H3K27me3 regulation to other chromatin marks, we found that the majority of non-promoter peaks of H3K27me3 do not overlap enhancers. These data raise the possibility that rather that functioning to control transcriptional regulatory elements directly, H3K27me3 might contribute to gene regulation through an alternate mechanisms such as by impacting chromatin architecture.”

2) Authors need to provide data in the cultures, or in vivo if relevant databases exist, for the presence/absence of the H3K4me3 modification on early and late promoters, as well as the H3K4me1mark for enhancers.

The data on H3K4me3 have been added to Figure 2A, B and are presented in the results on Page 9. Because we added this analysis of other chromatin features that co-localize with H3K27me3 at promoters, we also moved the data on H3K27ac and DNAse accessibility at H3K27me3-regulated promoters out of what was Figure 4D-E into what is now Figure 2 – supplement 1 at this point in the text. These data show that H3K4me3 levels (as well as H3K27ac and accessibility) inversely correlate over developmental time with H3K27me3 at promoters. These data support our conclusion that the regulation of H3K27me3 at gene promoters is associated with changes in transcription of the underlying gene.

They should also explain/speculate in the discussion why they think there is no phenotype in the KO mice with respect to cerebellar development, despite the findings on changes in the early/late genes?

We do not suggest that there is no phenotype in the cerebellar Kdm6b knockout mice. We state on Pg. 5 in the introduction, “We previously showed that loss of *Kdm6a/b* had no effect on postnatal morphogenesis of the cerebellum, demonstrating these enzymes are not required in vivo for regulation of CGN progenitor proliferation (Wijayatunge et al., 2018).” Our data here showing that the loss of Kdm6b predominantly impairs the expression of genes normally induced during maturation of postmitotic CGNs suggests instead that we might see defects in synaptic function in these mice. Notably, germline Kdm6b knockout mice die at birth because they fail to breathe properly, and this phenotype has been traced to failed maturation of a respiratory circuit in the midbrain. In the introduction on Page 4 we state, “Germline knockout of *Kdm6b* does not impair the gross architecture of the embryonic brain, but it does result in perinatal lethality caused by disruption of the functional maturation of a respiratory circuit that is required for proper breathing after birth (Burgold et al., 2012).”

We refer to these data again in the Discussion section on Page 24 and now relate our current findings to this phenotype, stating, “Indeed, a prior study showed that in a BAC transgene, enzymatically dead KDM6B was unable to rescue perinatal death in KDM6B knockout mice, which is consistent with a requirement for the enzymatic function of KDM6B in maturation of the respiratory circuit (Burgold et al., 2012). Our data now identify a set of genes induced in maturing neurons that function at synapses and require active H3K27me3 demethylation for their induction. Several of the targets of KDM6B-regulated demethylation and induction encode gene products known to play key roles in the functional maturation of synapses.”

We are currently studying the consequences of Kdm6b knockout for cerebellar synapse formation and function in our laboratory, but reporting those data is beyond the scope of the current manuscript.

If they have a more detailed time course of changes in late gene expression, when early genes are repressed persistently after treatment with EZH2 inhibitor, they should add this information, otherwise discuss any relevant information from previous studies bearing on this issue.

We have added additional time course data to the GSK-126 experiments in Figure 7G as the reviewer suggested. These data are reported in the Results section on Page 21.

To determine when inhibiting Ezh2 could result in impaired expression of the maturing CGN gene expression program, we added GSK-126 (or DMSO as control) at either DIV1 or DIV3 and then harvested CGNs at DIV5 for RNA-seq (Figure 7G). These data are important because they show that when we inhibit EZH2 at DIV3, which is after the main period when early gene promoters gain H3K27me3, we still impair late gene expression without significant induction of the early genes. The implications of these data are discussed on Pg. 26.

We have added the following experimental details about the drug treatments to the methods section under “CGN culture” on Page 28. “The KDM6 family inhibitor GSK-J4 (Σ #SML0701) (Kruidenier et al., 2012) and the EZH2-specific inhibitor GSK-126 (Cayman #15415) (McCabe et al., 2012) were first used in a dose response assay to determine IC_50_ in CGNs. For this, Cell Titer Glo (Promega #G7570) was performed on CGNs cultured in a 96-well plate at density of 100,000 cells/well, with inhibitors added one time at DIV1 and CGNs processed for downstream assays at DIV5. Further experiments with inhibitors were performed by treating CGNs one time at either DIV1 or DIV3 and processing for downstream assays at DIV5, using doses as specified in the text.”

We know that Ezh2 inhibition persist over the time course we use the drug in culture because we show in the western blot in Figure 7A that a single dose of GSK-126 delivered at DIV1 is sufficient to result in globally reduced H3K27me3 at DIV5.

We did not test if the effects of GSK-126 on H3K27me3 or gene expression can be reversed or maintained over longer periods of time because it is challenging to keep mouse CGNs alive in culture for more that 7-10 days. As to data from the literature, to our knowledge only one previous study has specifically examined the functions of EZH2 in postmitotic stages of neuronal differentiation (Buontempo et al., 2022) now cited on Pg. 27. This paper used conditional EZH2 knockout in cells that were differentiated from ESCs via neuronal progenitors to glutamate neurons in culture and it also did not address the later time points or reversibility.

3) We are not requesting new experiments related to autism, but if they choose to emphasize this point, authors are requested to discuss how their results with PRC2 provide insight into the epigenetic basis for autism, in the context of previously published studies by others on additional chromatin factors, as mentioned in the reviews.

We have revised or added text to the introduction on Page 5 and the discussion on Page 25 to add more information on the ASD relevance of KDM6B in the context of other ASD-associated epigenetic factors.

As the reviewers are well aware, there are a large number of chromatin regulators that have been implicated as ASD risk genes. To introduce our readers to this idea, and to place KDM6B in context, in the introduction on Page 5 we revised the text to state, “Chromatin regulators are one of the major classes of genes associated with autism spectrum disorder (ASD) and notably *KDM6B* has been identified as a high confidence ASD risk gene in humans (Satterstrom et al., 2020). *Kdm6b* haploinsufficiency in mice is associated with ASD-like impairments in sociability suggesting its importance for proper social network function in the adult brain (Gao et al., 2022), however the underlying chromatin and transcriptional mechanisms of these Kdm6b mutation phenotypes remain to be elucidated.”

On Pg. 25 we added text to discuss how our identification of KDM6B target genes may enhance understanding of the relationship between this gene and ASD risk, saying, “Several of the targets of KDM6B-regulated demethylation and induction encode gene products known to play key roles in the functional maturation of synapses. These include the glutamate receptor subunit genes *Grin2c* and *Grm4,* which play important roles in cerebellar motor function and synaptic plasticity (Kadotani et al., 1996; Pekhletski et al., 1996), the GABA-A receptor δ subunit gene (*Gabrd*), which modulates the subcellular localization and neurosteroid sensitivity of GABA-A receptors (Vicini et al., 2002) and the neuronal pentraxin receptor gene *Nptxr,* which promotes the formation and stabilization of both excitatory and inhibitory synapses (Lee et al., 2017). Impaired or delayed induction of these genes in the absence of KDM6B has implications for understanding how mutations in KDM6B may lead to behavioral phenotypes in ASD via altered maturation of neural circuit function.”

With respect to biological convergence between KDM6B and other ASD-associated chromatin regulators, a recent study showed that H3K27me3 changes may compensate in part for the loss of *Dnmt3a* as a mechanism of gene repression. On Page 25 we cite this work, saying, “Interestingly, mutations in another ASD-associated chromatin regulator, DNMT3A, that impair DNA methylation-dependent mechanisms of gene repression appear to be partially compensated by changes in H3K27me3 (Li et al., 2022). Future studies will be needed to address whether modulation of H3K27me3 could function as a point of biological convergence downstream of multiple ASD-associated chromatin regulators.”

As to potential links between enhancers and genes associated with ASD risk, which was referenced by one reviewer as mentioned in (Ramirez et al., 2022), it is our opinion that this an interesting area for future study, but well beyond the scope of what we have the space to discuss here.

Reviewer #1 (Recommendations for the authors):

We have paraphrased many of the reviewer’s comments here for brevity and grouped them into related questions.

1) Novelty of this work in the context of prior literature.

The vast majority of studies in the literature that have addressed functions of H3K27me3 focus on questions of fate commitment in dividing cells. We cite a sampling of those papers here. On Page 3 we state the major difference between the current study and those prior studies of H3K27me3 in development, saying, “Neurons are born very early in their overall lifespan, and they undergo significant transcriptional and functional changes during postmitotic stages of their developmental maturation. The enzymes that write and erase H3K27me3 remain expressed in neurons both during postnatal stages of development and in the adult brain (Wijayatunge et al., 2014; Wijayatunge et al., 2018), suggesting that they have functions beyond fate determination that remain to be understood.” We go on to identify one of these functions as orchestrating the program of mature gene expression that turns on over time in postmitotic CGNs, and we further demonstrate that both H3K27me3 demethylation mediated by KDM6B and H3K27me3 methylation, mediated by Ezh2, contribute to the fidelity of this process. We think the novelty of the work is clear.

2) The biological function of the chromatin regulatory pathways studied – the conclusion in the abstract that they have uncovered gene programs underlying functional neuronal maturation is a bit overstated because there are no functional studies in this work.

We agree that we have not demonstrated functional maturation of our neurons. However, we have demonstrated that the chromatin mechanisms studied here do regulate the expression of genes that are known to underlie functional maturation of synapses. One example is the developmental switch in NMDA-type glutamate receptor subunits from *Grin2b* to *Grin2c*, which is well established to underlie developmental changes in synaptic function in CGNs (see Kadotani et al., 1996).

To address the reviewer’s concern that we do not directly show maturation, we have thus changed the sentence paraphrased above from the abstract to read, “We find that expression of this maturation gene program in cerebellar granule neurons (CGNs) requires dynamic changes in the genomic distribution of histone H3 lysine 27 trimethylation (H3K27me3), demonstrating a function for this chromatin modification beyond its role in cell fate specification.”

3) The results and figures are written somewhat for a neurodevelopmental audience. Acronyms for the early and late genes are not defined, nor do they say how/why the representative genes were selected for certain experiments, or how the time course of events in culture relates to the timing in vivo, other than early or late.

We appreciate that the names of neurobiology-relevant genes may not always been obvious to non-neurobiologists. In addition, in any study that reports genome scale data it is challenging to provide details on every gene shown. To make the manuscript more accessible to all readers, we have spelled out the names and annotate the functional importance of key early (*Myc, Atoh1*) and late (*Grm4, Grin2c*) genes that appear repeatedly in the text and figures. We hope this will help the readability of the study.

To address the point about how the time course of events in culture relates to the timing in vivo, we have added a rank-rank hypergeometric overlap analysis of the gene expression changes in culture compared with the changes in vivo. This analysis allows for a global view of how two differential gene expression programs relate to one another. This plot has been added as Figure 6A, and the results are presented on Pg. 17.

4) Details of genes in the GO terms and comparison of GO terms between ChIP- and RNA-seq data for related experiments.

We have redone the GO analysis with an improved tool described in the methods. We list all the genes used for GO analysis in Table S3. The methods on Pg. 35 now reads, “Gene ontology analysis was performed using the GO consortium GO tool (Ashburner et al., 2000) using terms related to ‘Panther GO biological process slim’ (Mi et al., 2019). Genes used to generate GO terms are provided in Table S3.”

5) Scale indications missing for the ChIP and RNA-seq tracks.

We have added these scale markings to all figures. In every case the scale is the identical for all conditions being compared for a single type of data (e.g. H3K27me3 ChIP-seq at different ages or RNA-seq +/- drug treatment).

6) A statistical analysis section should be included in the methods. Figures often show large statistical differences with multiple *'s, but only "* indicate p< 0.05" in the legends.

Statistical methods are included in the methods section on Page 34, and additional detail on statistical significance represented by asterisks has been provided in each figure caption.

7) Authors should also consider another labeling system for comparing the levels of binding and expression in different contexts rather than up/down, gain/loss, up/down/gain/loss before and after treatment, particularly when referring to previous figures. Authors use "loss" and "gain" exclusively in the text, but in their figures, they use "up" and "down", which makes it difficult to track when they introduce the RNA-seq results.

We have revised the text. We appreciate that comparing multiple types of data in this story, as well as the fact that we are usually looking at variation in ChIP signal or gene expression over time, can make the figures complex to follow. We have done our best to synchronize the use of terms such as up and down or gain and loss across the figures and the text. We now refer to H3K27me3 peaks that are lost with the GSK-126 treatment as “sensitive” peaks.

Reviewer #2 (Recommendations for the authors):This study provides numerous well-controlled experiments to define the dynamics of H3K27me3 in cerebellum development. I have some additional comments on areas that could be addressed:– The analysis of the effects of KDM6B deletion on H3K27me3 and gene expression makes a convincing case that the activity of this enzyme is important for the removal of H3K27me3 and gene activation during postnatal maturation, but a direct assessment of the overlaps of the genes that are differentially expressed in development and those that are differentially expressed upon KDM6B deletion is not prominently presented. The degree to which there is overlap between these gene lists, and the identity of these genes could be helpful for understanding the functional importance of KDM6B and H3K27me3 removal during neuronal maturation.

We thank the reviewer for this suggestion. We have added gene expression data from P7, P14 and P60 cerebellum for the clusters in Figure 4 – supplement 2A that show genes upregulated and downregulated due to *Kdm6b*-cKO. Indeed, we see that genes downregulated due to *Kdm6b*-cKO are induced in mature neurons, while genes upregulated due to *Kdm6b*-cKO are repressed in mature neurons. We present the results on Pg. 13 where we added text saying, “Genes that were elevated at P14 in the *Kdm6b*-cKO mice all showed developmentally regulated expression in control mice that was highest at P7 and fell by P60. Genes whose expression was reduced in the *Kdm6b*-cKO mice also showed developmentally regulated expression, but they increased in expression over developmental time. Thus, the deletion of Kdm6b in GNPs appears to slow the transcriptional maturation of CGNs in postnatal life.”

– Although the analysis of H3K27me3 dynamics at promoters and gene bodies is informative, the authors do not provide any analysis of enhancers in the study. The study would be significantly strengthened if some assessment of H3K27me3 was carried out at these sites. Given that the authors have access to histone modification profiles that would allow them to identify putative enhancers in the cerebellum (H3K27ac Frank et al. 2015), it should be possible for the authors to carry out this analysis without completing additional experiments.

We have added these data as suggested. The changes are detailed in the Essential Revisions section above.

– The study takes advantage of the CGN culture system in order to do insightful pharmacology experiments. These cells show similar H3K27me3 dynamics as in vivo samples, making it a suitable system for their analyses. However, the authors do not make a direct comparison of the genes that have changes in H3K27me3 and gene expression in this system and the genes that show similar changes in vivo. A direct comparison of the genesets by overlap analysis would be helpful for the reader to know if the same genes are affected in both systems, or if the concept of dynamic H3K27me3 is the same, but the specific loci affected are context dependent.

We thank the reviewer for this suggestion. We have now added violin plots in Figure 6 – supplement 1E describing H3K27me3 enrichment from P7 to P60 as a function of genes within clusters defined by differential H3K27me3 levels in culture, from Figure 6D (previously Figure 5D). These data show concordant regulation of the same genomic elements in the two system. Presentation of these data have been added to the Results section on Pg. 19.

– It would be helpful if the authors could better describe how the clusters in Figure 2A were defined. For example, the dendrogram could be included and show the remaining genes, not in the clusters presented. For many "fast" and "slow" peaks the profiles look quite similar. It appears that differences between these clusters in terms of associated gene expression might be more distinct if the slow population was more restricted to the genes at the bottom of that cluster that displays much more clear persistence of methylation through P14.

We have now added the dendrograms used to hierarchically cluster the peaks in Figure 2A, 3B and 6D. As the reviewer suggests, the dendrogram shows these peaks could be broken down into additional clusters, however this may be at the expense of biologically relevant information.

– The quantitative analysis of DHS in Figure 4E (left) shows a reduction in DHS signal at P7 to P60 "H3K27me3 up" promoters but the aggregate plot (right) appears to show an increase in DHS signal at these sites. No mention of these contradictory results is made in the text. Do the authors have an explanation for this disparity?

In what is now Figure 2 – supplement 1 (formerly Figure 4E), we show a distribution of log2 fold changes in H3K27ac and DHS signal between P60 and P7, as a function of genes belong to clusters H3K27me3 Down and H3K27me3 Up. While H3K27me3 Down promoters are associated with a significant increase in DHS signal, and conversely H3K27me3 Up promoters are associated with a significant decrease, the overall log2 fold change signal from P7 to P60 is above 0 in all comparisons, suggesting an overall increase in chromatin accessibility at promoters. So, while the metagene plot on the right shows an increase in DHS signal at P60, this increase is significantly less compared to genes at H3K27me3 Down genes as well as genes with Unchanged H3K27me3 at promoters. We hope this clarifies the data. We felt these data were not as important as some others in the manuscript, thus we moved them to the supplementary files.

– The gene ontology analysis that is performed in the study does a convincing job of showing the general classes of molecules affected by H3K27me3 dynamics, but very limited analysis or discussion is made of specific genes that are dysregulated and how they may functionally influence CGN maturation. While the characterization of the functional effects of disrupted expression for these genes may be beyond the scope of this manuscript, the study would be improved if a few examples were noted and discussed.

We have added a section to the discussion on Pg. 25 where we discuss the biological functions of some of the most interesting KDM6B-dependent late genes we identified, *Grin2c*, *Grm4*, *Gabrd*, and *Nptxr.*